# Current and Expected Trends for the Marine Chitin/Chitosan and Collagen Value Chains

**DOI:** 10.3390/md21120605

**Published:** 2023-11-23

**Authors:** Helena Vieira, Gonçalo Moura Lestre, Runar Gjerp Solstad, Ana Elisa Cabral, Anabela Botelho, Carlos Helbig, Daniela Coppola, Donatella de Pascale, Johan Robbens, Katleen Raes, Kjersti Lian, Kyriaki Tsirtsidou, Miguel C. Leal, Nathalie Scheers, Ricardo Calado, Sofia Corticeiro, Stefan Rasche, Themistoklis Altintzoglou, Yang Zou, Ana I. Lillebø

**Affiliations:** 1CESAM—Centre for Environmental and Marine Studies, Department of Environment and Planning, Campus Universitário de Santiago, University of Aveiro, 3810-193 Aveiro, Portugal; helena.vieira@ua.pt (H.V.); goncalo.lestre@ua.pt (G.M.L.); sofiacorticeiro@ua.pt (S.C.); 2Nofima Norwegian Institute of Food Fisheries and Aquaculture Research, Muninbakken 9-13, 9019 Tromsø, Norway; runar.gjerp.solstad@nofima.no (R.G.S.); kjersti.lian@Nofima.no (K.L.); themis.altintzoglou@Nofima.no (T.A.); 3ECOMARE, CESAM—Centre for Environmental and Marine Studies, Department of Biology, Santiago University Campus, University of Aveiro, 3810-193 Aveiro, Portugal; anacabral@ua.pt (A.E.C.); miguelcleal@ua.pt (M.C.L.); rjcalado@ua.pt (R.C.); 4GOVCOPP—Research Unit on Governance, Competitiveness and Public Policies, DEGEIT, Campus Universitário de Santiago, University of Aveiro, 3810-193 Aveiro, Portugal; anabela.botelho@ua.pt; 5Fraunhofer Institute for Molecular Biology and Applied Ecology IME, Forckenbeckstrasse 6, 52074 Aachen, Germany; carlos.helbig@ime.fraunhofer.de (C.H.); stefan.rasche@ime.fraunhofer.de (S.R.); 6Department of Ecosustainable Marine Biotechnology, Stazione Zoologica Anton Dohrn, Via Ammiraglio Ferdinando Acton 55, 80133 Napoli, Italy; daniela.coppola@szn.it (D.C.); donatella.depascale@szn.it (D.d.P.); 7Flanders Research Institute for Agriculture, Fisheries and Food, ILVO, Aquatic Environment and Quality, Jacobsenstraat 1, 8400 Ostend, Belgium; johan.robbens@ilvo.vlaanderen.be (J.R.); kyriaki.tsirtsidou@ilvo.vlaanderen.be (K.T.); 8Research Unit VEG-i-TEC, Department of Food Technology, Safety and Health, Ghent University Campus Kortrijk, Graaf Karel de Goedelaan 5, 8500 Kortrijk, Belgium; katleen.raes@ugent.be (K.R.); yang.zou@ugent.be (Y.Z.); 9Department of Life Sciences, Chalmers University of Technology, 412 96 Göteborg, Sweden; nathalie.scheers@chalmers.se

**Keywords:** collagen, chitin, chitosan, circular (bio)economy, market opportunities, sustainability, SWOT, PESTEL

## Abstract

Chitin/chitosan and collagen are two of the most important bioactive compounds, with applications in the pharmaceutical, veterinary, nutraceutical, cosmetic, biomaterials, and other industries. When extracted from non-edible parts of fish and shellfish, by-catches, and invasive species, their use contributes to a more sustainable and circular economy. The present article reviews the scientific knowledge and publication trends along the marine chitin/chitosan and collagen value chains and assesses how researchers, industry players, and end-users can bridge the gap between scientific understanding and industrial applications. Overall, research on chitin/chitosan remains focused on the compound itself rather than its market applications. Still, chitin/chitosan use is expected to increase in food and biomedical applications, while that of collagen is expected to increase in biomedical, cosmetic, pharmaceutical, and nutritional applications. Sustainable practices, such as the reuse of waste materials, contribute to strengthen both value chains; the identified weaknesses include the lack of studies considering market trends, social sustainability, and profitability, as well as insufficient examination of intellectual property rights. Government regulations, market demand, consumer preferences, technological advancements, environmental challenges, and legal frameworks play significant roles in shaping both value chains. Addressing these factors is crucial for seizing opportunities, fostering sustainability, complying with regulations, and maintaining competitiveness in these constantly evolving value chains.

## 1. Introduction

The ocean represents ca. 95% of the biosphere and is crucial for the planet and humankind, as it provides a plethora of important resources and services [1,2]. Although currently recognized as a common provider of social, environmental, and economic benefits [3], the ocean has faced, and continues to face, several natural and anthropogenic threats. Some of the major threats are related to the overexploitation of marine resources, climate change, pollution, ocean acidification, habitat damage, and management failure [4]. To mitigate these major threats, it is critical to maintain the balance between the exploitation of marine resources and the ecosystem resilience to such exploitation. This balance should be evident to all, as well as coordinated with and integrated into public policies, governance, finance, and management of global supply chains where ocean resources play a role [3]. To achieve this goal, sustainable and circular business models, as well as integrated policies that protect marine ecosystem functions and regulate all major activities occurring in the ocean, must be implemented or improved across the globe.

The blue economy reached a Gross Value Added (GVA) of EUR 129.1 billion and a turnover of EUR 523 billion in 2020 across seven different sectors (living resources, non-living resources, marine energy, port activities, shipbuilding and repair, maritime transport, and coastal tourism) [5]. The marine living resources sector comprises the harvesting and farming of biological resources, as well as their conversion and distribution, and that sector alone generated more than EUR 19.4 billion in GVA and EUR 119 billion in turnover in 2020. Despite this GVA, it may still be underestimating the value of the EU blue bioeconomy as a whole, as this does not encompass sectors such as blue biotechnology. Fisheries and aquaculture, two ocean-related major economic activities, have grown throughout the years, in part due to the increasing demand for food by an expanding human population [6]. If exploited sustainably, ocean resources potentially have the capacity to regenerate and feed a large proportion of the world’s population. According to the Food and Agriculture Organization of the United Nations (FAO), aquaculture accounted for 122.6 million tonnes of the unprecedented total of 214 million tonnes produced by fisheries and aquaculture in 2020 [6]. In this year, the number of people employed in primary fisheries and aquaculture exceeded 58 million [6], indicating the importance of these activities in the economic development of multiple countries. Moreover, as marine organisms have evolved for thousands of years to be able to thrive in complex habitats and are exposed to extreme conditions, they produce a wide variety of specific and potent bioactive substances [7]. Hence, the ocean is a rich and natural source of many bioactive compounds that cannot be found elsewhere. Thousands of marine bioactive compounds have been extracted, identified, and characterized in recent decades [8]. Indeed, ~7000 of these molecules are already in use or being validated for several purposes, ranging from medicine to industrial applications [9]. For instance, in 2020 and 2021, 1407 and 1425 new bioactive compounds were reported from marine organisms [10]. However, the increased extraction and use of such compounds has been exerting even more pressure on the limited natural resources of the marine realm.

Environmental and economic concerns have been increasingly driving the use of eco-friendly alternatives to exploit marine natural resources. In the age of sustainability, where development models are changing towards circularity and zero waste, the fisheries and aquaculture sectors, alongside many of the other sectors they connect with (like fish and seafood transformation industries), are key players in supplying new by- and co-products that work as raw materials for other industries. Examples include the once considered “waste streams” of fish by-catches, the shells and non-edible parts of shellfish and crustaceans, and invasive species such as crabs and starfish, which can serve as raw materials for different bio-based products. Many industries, including the pharmaceutical, veterinary, nutraceutical, cosmetic, biomaterials, and others, benefit from the development of products and/or processes using these marine resources [11]. Such products may take the form of pharmaceutical drugs, livestock feed formulas, pet food products, specialty foods and nutritional supplements for several human conditions, medical biocomponents, beauty supplements, functional textiles or new fibres, biomaterials used in construction or nature-based building solutions, and additives or enzymes used in manufacturing and industrial processes, just to name a few, to improve productivity with lower environmental impacts [12,13,14,15]. These approaches promote the development of sustainable products, circular (bio)economy models, zero-waste strategies, and reduce environmental pollution.

Chitin, its derivative chitosan, and collagen, are highly relevant marine bioactive compounds to the biomedical, nutraceutical, cosmetic, feed, and wastewater treatment industries, among others [12,16,17,18,19]. Both chitin and collagen represent unified templates for biomineralization and skeletogenesis in many organisms and are essential elements for their structural life support functions [20]. In fact, both biopolymers represent examples of the “scaffolding strategy”, a modern trend of using naturally occurring 3D scaffolds made of chitin and collagen (i.e., in sponges) for tissue engineering and technology derived thereof [21,22,23]. These naturally occurring compounds, or derivatives, are also used in applications such as preservative food coatings due to their thermal stability and antimicrobial qualities [24] but also in a wide range of different biomaterials, some even in the framework of extreme biomimetics inspiration [25,26].

Chitin is one of the most abundant biopolymers in nature [27]. It can be extracted from the exoskeletons of crustaceans, molluscs, insects, and fungi. It can also be obtained from some *Porifera*, like sponges [28]. Chitin is classified in three different groups: α-chitin, usually extracted from the exoskeleton of crustaceans such as shrimps and crabs; β-chitin, extracted from squid pens; and γ-chitin, obtained from fungi and yeasts [29]. Chitin and chitosan properties are highly variable depending on their source, as well as on the deacetylation, protein concentration, and extraction procedures [30]. The conventional way of making chitin and chitosan include demineralisation, deproteinisation (+deacetylation for chitosan), or electrochemical methods [31]. Both chitin and chitosan undergo modifications (e.g., deacetylation, quaternization, oxidation) to enhance their physical properties [32]. Although chitin has poor solubility, its derivative chitosan is a soluble biopolymer in aqueous acidic conditions [33]. Therefore, chitin is often chemically modified by deacetylation to obtain chitosan.

Collagen has at least 28 types (I-XXVIII) described. The most abundant types are in mammals, fibrillar collagen types I–III, predominantly sourced from commercialized porcine, bovine, ovine, and chicken tissues [34]. It can also be obtained from marine sponges [35,36], jellyfish, squids, and fishes [37]. The skin, bones, fins, head, and scales of fish are rich in collagen and account for approximately 75% of the fish wet weight [38]. Collagen has multiple sources, but an increase in marine-derived collagen is being seen [39,40] and its usages range from cosmetic and nutraceutical preparations to tissue engineering, medical or pharmaceutical high-value products [41,42], and even several manufacturing biomaterials applications [43,44]. In fact, collagen from marine organisms utilised for biomedical applications has been recognised as a convenient and safe source, and some advantages have been pointed out when compared to collagen from mammalian origin, including (1) less significant religious and ethical constraints; (2) greater absorption due to low molecular weight; (3) low inflammatory response; (4) and minor regulatory and quality control problem [45]. Even more, it represents an option towards the valorisation of marine by-products and the development of the circular economy concept, as providing new solutions for the reuse of materials is highly targeted on the EU policy making agenda [46].

As chitin and collagen can be extracted from sources that would otherwise be considered as waste (e.g., non-edible parts of fish and shellfish, fisheries’ by-catch, and invasive species), the use of these compounds represents an opportunity to reinforce circular business models and to reuse and reduce the waste streams derived from marine fisheries, aquaculture, and food processing industries. Chitin and collagen markets currently represent USD ~7900 million and USD 4700 million, respectively [47], meaning they both have substantial commercial interest. The application and transformation of what was once considered waste has therefore led to new valorisation strategies, creating opportunities to capitalize these co-products and side streams in market segments not yet explored [12,48], building novel business models in new value networks for the marine-derived chitin/chitosan and collagen.

In this view, a systematic scientific literature review was performed in the present study to address the following:The extent of scientific knowledge along the marine-derived chitin/chitosan and collagen value chains.How stakeholders should interact within each value chain to narrow the gap between scientific knowledge on chitin/chitosan and collagen and their industrial application.

Although the concept of “value-chain” is evolving to “value-network”/ ”value webs” [49], the present study still uses “value-chains” for simplifying the first approach to this subject. 

Eight drivers of change [50] were considered for developing the marine-derived chitin/chitosan and collagen value chains analysed here: (1) raw material origin; (2) inputs/feedstock; (3) pre-treatment/pre-processing; (4) processing and product manufacturing; (5) standardisation/certification; (6) packaging/distribution; (7) consumption; and (8) value chain outputs. Mapping the involved stakeholders allowed us to identify the main sectors that explore the marine sources of chitin/chitosan and collagen. As for the remaining drivers, the intervening players were identified from the literature on marine chitin/chitosan [51,52,53] and collagen [52,54] production processes and on product valorisation and applications.

The results are discussed considering the following: (i) the research effort on the initial stages of the marine-derived chitin/chitosan and collagen value chains; (ii) sources’ sustainability, following social and environmental standards; (iii) how market trends may influence the development of new products and applications for these compounds and their derivatives and the business model, focusing on the principle of circular economy to prevent/reduce waste [55]. To evaluate the current state and expected trends for the marine-derived chitin/chitosan and collagen value chains, a strengths, weaknesses, opportunities, and threats (SWOT) analysis was applied, followed by a political, economic, social, technological, environmental, and legal (PESTEL) analysis. The results of these analyses are discussed considering the characteristics and evolution of the aquaculture and fisheries sectors during the last 70 years at the global level.

## 2. Results

### 2.1. Trends in the Distribution and Number of Publications per Value Chain

The number of peer-reviewed publications (hereafter referred to as publications) related to the chitin/chitosan value chain was almost twice that of publications related to the collagen value chain (138 vs. 84). Four publications contained information relevant for both value chains. Approximately half of the analysed publications were published in top tier (i.e., Q1) journals. For the chitin/chitosan value chain, 49% of the publications analysed (*n* = 67) were published in journals in Q1 and 31% (*n* = 43) in Q2. For the collagen value chain, 50% of the publications (*n* = 42) were published in journals in Q1 and 35% (*n* = 29) in Q2. Globally, for both value chains, publications were distributed as follows: Q1, 48% (*n* = 106); Q2, 32% (*n* = 71); Q3, 12% (*n* = 26); and Q4, 7% (*n* = 16).

As for the evolution of the number of publications related to each value chain (Figure 1), the first scientific publication approaching the chitin/chitosan value chain was published in 1990, a second in 1992, and a third in 1993 (Figure 1a). After a 7-year gap, a fourth publication was published in 2000; after a period of intermittent publication from 2001 to 2009, publications related to the chitin/chitosan value chain have been published yearly, with an increasing trend being recorded over the years (Figure 1a). The maximum number of publications (*n* = 28) was observed in 2022, with 20 being published in Q1 journals.

The first scientific publication approaching the collagen value chain was published in 1969, followed by a second and third publication in 1971 and 1972, respectively, and a fourth and fifth in 1994 and 2000 (Figure 1b). After a 5-year gap, a publication was published in 2006, but only since 2009 have publications been published on this topic on a yearly basis. An increasing trend has been observed since 2009 (Figure 1b), with the maximum number of publications in 2022. In this year, 12 of the 21 publications were published in Q1 journals.

### 2.2. Trends in the Geographical Origin of Publications per Value Chain

The scientific publications related to each value chain were differently distributed based on the country of the corresponding author(s). Publications related to the chitin/chitosan value chain originated from 43 countries (Figure 2), whereas those related to the collagen value- chain originated from 25 countries (Figure 3). Most publications related to the chitin/chitosan value chain were from India (n = 20), while most publications related to the collagen value chain originated from China (n = 13), closely followed by India (n = 12). Asia was the most relevant region, with 43% and 57% of the corresponding authors of publications related to the chitin/chitosan and collagen value chains, respectively, being based in Asian countries.

### 2.3. Trends in the Origin of the Marine Raw Materials and Feedstock per Value Chain

The origin of the raw material(s) used differed considerably between the two value chains, based on the information provided by the analysed publications (Figure 4). For the chitin/chitosan value chain, the “food processing industry” and “fisheries” were the most frequent sources of raw materials used in publications (34% and 31%, respectively) (Figure 4a). The source “aquaculture” showed a low value (6%), considering the rising interest in this sector related to the aquaculture production of species that may be a source of chitin and its derivatives, such as chitosan (i.e., crustaceans) [6]. For the collagen value chain, most publications used raw materials from “fisheries” (52%) followed by the “food processing industry” (22%) (Figure 4b). Although “aquaculture” was also the least frequent source of raw materials in collagen value chain publications, its relative contribution was twice that calculated for the chitin/chitosan value chain (12% vs. 6%, respectively). Globally, “fisheries” have been the most relevant source of raw materials for both value chains. It is worth noting that “undisclosed” was the third most common source on both value chains; furthermore, in the chitin/chitosan value chain, its value (29%) was similar to that of the two most common sources (Figure 4a).

For the chitin/chitosan value chain, “crustacean waste” was the most used feedstock in the studies analysed (71%), especially “shrimp waste” (35%) (Figure 5a). The percentage obtained for “algae and seagrasses” (15%) resulted from a single publication that mentioned endophytic fungi isolated from 19 different species of algae and 10 different species of seagrasses [56]. Regarding the collagen value chain, “fish scales, skin, and bones” were the feedstock used in 62% of the analysed publications (Figure 5b). Globally, fish and crustacean wastes were the most used feedstock in the studies related to both value chains.

### 2.4. Trends in the Perception of Sustainability for Chitin/Chitosan and Collagen Value Chains

The sustainability, as expressed in the scientific publications, for each value chain was categorized into economic, environmental, and social. Economic sustainability is mostly related to the improved cost efficiency of the extraction methods, especially regarding them being cheaper than previously established methods or capable of achieving a higher quality or higher quantity of compounds. Environmental sustainability is related to environmentally friendly methods of compound extraction and to waste reduction and reuse. Social sustainability refers to practices that may improve society well-being and reduce inequalities, such as those related to consumer cultural or dietary needs.

Overall, more economic, environmental, and social sustainability practices have been applied in the chitin/chitosan value chain than in the collagen value chain, particularly environmental and economic sustainability practices (Figure 6). Environmental practices are the most referred to in publications related to both value chains, such as environmentally friendly methods of extraction [57,58], reduce/reuse of waste [59,60,61], or reduction in environmental harm [62], followed by economical practices, such as cheaper consumables [57,63], cheaper methodologies [64,65,66,67], more cost-efficient processes [68,69,70,71,72], and new potential products [19,73,74].

### 2.5. Trends in Market Applications for Each Value Chain

Regarding the market applications of chitin/chitosan and collagen, several different sectors were mentioned as both present and future applications. Overall, collagen products are currently less used than chitin/chitosan products (Figure 7), and an increased use of both types of products is expected, as described by the authors of the screened publications. Chitin/chitosan products are mostly used in the industrial sector, newly derived and purified compounds, food applications, and wastewater treatment (Figure 7) [56,71,75,76,77,78,79,80,81,82,83,84,85,86,87,88,89,90,91,92,93,94]. An increased use in these sectors, as well as in biomedical applications [95,96,97,98], is envisioned. However, in the analysed scientific publication, the authors state that more time is needed to assess how the use of chitin/chitosan products in biomedical applications will evolve [99,100]. Collagen products are mostly used in biomedical applications [68,101] and as purified compounds (Figure 7) [68,102], and a substantial rise in biomedical, cosmetic, and pharmaceutical applications is suggested by the authors in many of the analysed publications [37,61,62,103,104].

### 2.6. Trends in Data Distribution per Category of Information per Value Chain

There is a high degree of information discrepancy between the different categories of information presented in Figure 8, with more information presented in the categories relating to raw material origin, feedstock, pre-processing, and processing. While the sources and processes for obtaining chitin/chitosan or collagen were documented in >70% of the publications related to each value chain, market information related to the current applicability of both products and their derivatives was scarce (~23% in the case of the chitin/chitosan value chain and ~20% for the collagen value chain) (Figure 8). Moreover, the applicability of these products is generally documented as a possibility rather than a reality, and very few publications have mentioned patents, profitability, or marketability. Even when considering future perspectives, ~60% of the publications refer to products but only ~5% refer to market growth or profitability.

Although the economic and environmental sustainability of the chitin/chitosan value chain has been addressed in ~40% of the analysed publications, this value was much higher than that found for the collagen value chain (~30% for environmental sustainability and ~20% for economic sustainability) (Figure 8). Social sustainability was only seldom referred to for both value chains (<10% of publications). 

## 3. Discussion

### 3.1. Trends in the Distribution and Number of Publications per Value Chain

The present study shows increasing trends in the number of scientific publications related to the marine-derived chitin/chitosan and collagen value chains, particularly since 2010 (Figure 1). Despite this similar trend, the number of publications focusing on the chitin/chitosan value chain was higher than that focusing on the collagen value chain, even though the first publication for collagen was authored 20 years before that first addressing chitin/chitosan. This might be related to better knowledge on the range of properties and applications of chitin/chitosan products in various industries (e.g., agriculture, food, healthcare, textile), whereas many of the properties and applications of collagen are still being investigated and developed [105]. Although many scientific publications on chitin/chitosan (>10,000) have been published between 2000 and 2021 [106], very few (529 publications when combining our results for both Scopus and WoS databases) considered the chitin/chitosan value chain or presented a consistent market analysis for products based on these bioactive compounds [29,107,108,109,110,111]. Nevertheless, the potential of these compounds recognised in scientific research has been translated into commercial applications, with the markets of chitin/chitosan and collagen products being valued at USD 7900 and 4700 million, respectively, and growing ~5% each year [47,112]. Chitosan is expected to reach a record compound annual growth rate (CAGR) of 17.3% between 2022 and 2030, reaching a market value of USD 15,100 million. It is important to underline the role that the USA and China have in the global chitin and chitosan derivatives market. The USA have a market estimated at USD 2300 million, and China is forecasted to reach a market size of USD 4100 million by 2030. Countries such as Japan, Canada, and Germany are expected to grow above the average rate, with a CAGR of 14.1%, 12.9%, and 10.7%, respectively. The marine collagen market size was valued at USD 1100 million in 2022, with an expected CAGR of 9.5% for the following ten years. Based on product type, gelatin products reached a market size of USD 633 million in 2022, while native collagen accounted for 25% of the market revenue share, with the different types of modified collagen accounting for the other 75%, due to the target of specific consumer needs and demands [113]. Moreover, the number of patents granted to chitin/chitosan derivatives in the European Union (EU) and the United States of America (USA) has increased throughout the years, along with the budget for research grants on these products [12]. Thus, although the bias in the number of scientific publications towards the initial steps of the value chain seems not to compromise the later steps of the value chain (e.g., marketing and patenting), the use of marine-derived chitin/chitosan and collagen products might be slower than that anticipated by their potential [114]. This might be reflective of the lack of publicly available data and scientific knowledge of the intermediate stages of product development. A closer collaboration between researchers interested in chitin/chitosan or collagen and potential end-users and industry players, focused on commercial viability and market fit, should therefore be encouraged from the early stages of research on these compounds.

### 3.2. Trends in the Geographical Origin of Publications per Value Chain

Researchers based in China and India have been the major contributors to the scientific knowledge on the marine-derived chitin/chitosan and collagen value chains, which agrees with both countries ranking among the top five producers of scientific and citable documents (https://www.scimagojr.com/countryrank.php, accessed on 1 February 2023). This is in line with data published also for the chitin/chitosan and collagen patents geographical coverage, with China being the top patenting country (45% and 56% of described patent families, respectively), followed by the USA (14% and 10%, respectively), demonstrating the enormous academic and commercial interest of Asia in these value chains [115,116]. Moreover, the contributions of China and India may be because these countries lead the aquaculture and fisheries production worldwide [6], and both activities provide commonly used sources for raw chitin/chitosan and collagen materials [117,118]. China and India both have vast coastlines allowing access to an extensive variety of marine resources, including fishes, crustaceans, molluscs, and seaweeds, from which chitin/chitosan and collagen products can be derived. Furthermore, aquaculture production has nearly doubled from 2010 to 2020 [6], and this large increase in the production of fishes and crustaceans means that a larger pool of wastes, such as fish skins and crustacean shells, can be used for the production of marine-derived chitin/chitosan and collagen products. Considering the growing interest in developing sustainable and eco-friendly products based on these compounds [38,106,119], the top-level scientific expertise and easy access to raw materials in both China and India might explain why the highest number of publications related to the chitin/chitosan and collagen value chains examined in the present study were found in these countries. The EU and USA should take these facts into consideration if they wish to equal the levels of dedicated research performed in these two Asian countries; incentivizing and funding more research and development (R&D) and proof-of-concept projects, and fostering academia/industry R&D joint projects, might be interesting routes to maintain a high level of scientific investigation related to marine-derived chitin/chitosan and collagen.

### 3.3. Trends in the Origin of the Marine Raw Materials and Feedstock per Value Chain

The marine-derived chitin/chitosan and collagen value chains are highly relevant in terms of sustainability and the circular economy [120,121]. This is because the raw materials for producing marine-derived chitin/chitosan and collagen can be obtained from waste streams of the aquaculture, fisheries, and seafood processing industries. As the use of such wastes is expected to increase globally in upcoming years, driven by zero-waste policies and an increasing demand for more eco-friendly processes and products, the origin of raw materials and the impact of their increasing usage on marine ecosystems are becoming subjects of concern.

Based on the publications analysed in the present study, 31% of the raw materials used for extracting chitin/chitosan and 52% of the raw materials used for extracting collagen originated from fisheries (Figure 4). However, none of the publications revealed if the raw materials were obtained directly from fisheries or from their discards and/or by-catches. While using the latter as raw materials not only contributes to reducing waste but also provides or increases the economic value of these otherwise neglected products, obtaining raw materials via fishing campaigns targeted for that purpose raises concerns about overfishing and marine ecosystems’ degradation [6].

The seafood processing industry is another major source of chitin/chitosan and collagen raw materials as it provides crustacean and fish wastes as sustainable sources [40,122]. Although the use of such waste materials has raised concerns about food safety and food quality [123], mostly due to contamination by metals, antibiotics, or other chemicals, they represent 34% and 22% of the chitin/chitosan and collagen raw materials, according to the results of the present study. As expected from the compositions of crustacean shells (15–40% chitin; [124]) and fish skin and bones (40–50% collagen; [125]), crustacean wastes were the most frequent source of chitin/chitosan (71%) and fish wastes sourced most of the marine-derived collagen (62%). The use of seafood wastes as sources of chitin/chitosan and collagen is expected to increase even further, given the high content of such compounds in these otherwise undervalued sources, their high market values [47], and the growing concern for reducing the economic and environmental impact of seafood wastes [122].

Aquaculture has been identified as a potential source of raw materials for the production of collagen and chitin/chitosan [126,127]. However, the present analysis revealed that the contribution of aquaculture-sourced raw materials has been relatively low in both the chitin/chitosan (6%) and collagen (12%) value chains. Although these values are expected to increase with the projected increase in aquaculture production and the preference towards sustainable production processes [128], the low percentages obtained here may reflect the common use of farmed fish and shellfish wastes as ingredients for animal feeds or their burial or burn [40], even though this is a lower market value usage of such side streams as compared with other non-feed applications. Given that aquaculture production is mostly located in countries with low income (i.e., Bangladesh, India, Malaysia) [129], the conversion of fish and shellfish wastes into bone char for water purification, feedstock ingredients, and energy sources might be preferred [130] to chitin/chitosan and collagen extraction processes. To respond to food safety and quality concerns arising from using fish and shellfish wastes [123], the quality of raw materials from aquaculture must be strictly controlled, and there is a market-based growing demand for certified, sustainable, and responsible products [129]. Furthermore, choosing other applications for these aquaculture side streams might also contribute to raising the market value caption of such enterprises and contribute to more sustainable and circular value chains.

In this analysis, the high contribution of raw materials from an “undisclosed” origin to both chitin/chitosan and collagen products (29% and 14%, respectively) is of concern. Moreover, the lack of transparency of such practice, which can pose a risk for human health [123], and the inability to accurately trace the origin of raw materials in scientific publications also limits the reproducibility of results. Scientific research relies on accurate and transparent reporting of the methods and materials used. If the source of raw materials is not disclosed, it impairs the replication of experiments and constrains scientific progress. In addition, identifying the correct taxonomic classification of species and reporting the complete scientific name (and how it evolved in case of reclassification) is paramount for traceability and reproducibility. Moreover, using raw materials of undisclosed origin raises relevant questions on whether such materials complied with the current biodiversity, legal, and social frameworks desired for a sustainable development. The scientific community should set the example for what regards best practices and consensual choices and, therefore, it is within this community that the highest standards must be enforced. In line with the objectives of the *Nagoya Protocol* (which entered into force in October 2014 and is currently signed by 92 parties; https://www.cbd.int/abs/doc/protocol/nagoya-protocol-en.pdf accessed on 1 February 2023) and the *Biological Diversity Act* (published in 2002 by the Government of India; https://faolex.fao.org/docs/pdf/ind40698.pdf accessed on 1 February 2023), publications which do not disclose the origin of bioresources should not be accepted to guarantee a transparent and just usage of such resources. A transparent and traceable supply chain for marine-derived biomaterials, including chitin/chitosan and collagen, should be implemented following the *Nagoya Protocol*, thereby promoting the sustainable use of marine resources and ensuring the equitable distribution of benefits among the stakeholders involved [131]. The growing use of blockchain methods to enforce traceability, along with biomolecular and geochemical traceability methods being widely implemented [132], will also put additional pressure on this need.

### 3.4. Trends in the Sustainability of Each Value Chain

The chitin/chitosan and collagen value chains are contributing to a more sustainable and circular economy where waste is minimised by efficiently using other industries’ side streams. The present study highlights a higher number of publications mentioning any of the three categories of sustainable practices (environmental, economic, social) in the chitin/chitosan value chain than in the collagen value chain. This suggests that the chitin/chitosan value chain is implementing more sustainability practices than the collagen value chain, particularly economic and environmental sustainability practices. However, our analysis was based solely on the number of publications that discuss sustainability and therefore did not consider grey literature nor thoroughly compare the industrial practices of the two value chains. Hence, the results may not accurately reflect the actual sustainability of chitin/chitosan and collagen products, processes, and value chains. Moreover, the differences in the number of publications mentioning “sustainability” between the two value chains might be due to chitin/chitosan being used in a wider range of applications and for longer than collagen.

In both value chains, the economic sustainability mentioned in the analysed publications was related to new and more cost-efficient processes, using cheaper consumables and/or methodologies. An increased number of such processes has been confirmed in a previous study[133]. Using more cost-efficient practices, such as reducing the values invested per yield of final product (i.e., alternatives to expensive enzymes or equipment [71,134,135], processes that reduce the time of extraction [69,136], or less energy-demanding processes [137], which therefore reduce processing costs), or higher quality (and therefore more economically valued) products [72] was also referred to in the publications analysed for both value chains. Economic sustainability can achieve even higher standards if the transition from R&D to industrial application becomes more articulate and fluid in the countries that perform most of the R&D, as is the case of India [114]. However, a possible imbalance due to the lack of technology required to extract and process chitin/chitosan and collagen in countries where R&D is being developed and/or market demand is expanding is likely to negatively affect the economic sustainability of both value chains.

Social sustainability is heavily linked with social equity and equality. Despite the extremely low percentage of publications mentioning social sustainability practices, marine-derived chitin/chitosan, collagen, and their derivatives are likely to directly improve social equity for the producers and suppliers of raw materials. By providing employment opportunities for people in coastal communities, particularly in regions where fishing and seafood processing are major economic activities, and exploring new product and market routes, the chitin/chitosan and collagen value chains have the potential to achieve higher levels of social sustainability. In addition, such products may also improve social equality regarding the religious and/or cultural aspects of consumer choices (e.g., halal or non-mammal origin) [103,138].

The environmental sustainability mentioned in the publications analysed was mostly linked with the introduction of circularity principles in both value chains, namely, with the reuse of what would otherwise be considered waste in seafood-related industries. For the chitin/chitosan value chain, many publications referred to environmentally friendly methods of extraction, most of them using less chemicals and thus leading to less pollution. Regrettably, none of the publications mentioned how sourcing chitin/chitosan and collagen in some countries while processing them in others effects the environmental sustainability of both value chains, for example regarding carbon footprint assessments.

### 3.5. Trends in Market Applications for Each Value Chain

The chitin/chitosan derivatives market is currently worth USD 7.900 million in 2023 and is forecasted to reach USD 24.900 million by 2030, as it is growing at a compound annual growth rate (CAGR) of 15.3% [112]. The global collagen market, valued at USD 4.700 million in 2023, is expected to reach USD 7.200 million by 2030, following a CAGR of 5.3% [47]. These values reflect the vast applications of both chitin/chitosan- and collagen-derived products. Such a range of applications was also mentioned in the publications analysed in the present study, together with some prospective applications of both products and their derivatives in different industries.

Our analyses revealed that chitin/chitosan-derived products are historically used in industrial (highest number of mentions), food, water treatment, cosmetic, pharmaceutical, animal supplement, and biomedical applications (Figure 7), although at very low levels (<10 mentions in all sectors). Interestingly, the number of publications mentioning “new derived compounds” and a “purified compound” was higher than that found for most of the other categories, indicating that research on chitin/chitosan is still highly focused on the compound itself rather than on its market application and reflecting the previously identified slow transition from scientific research to commercial applications [114]. Unfortunately, this is not expected to change soon, with the number of mentions for “new derived compounds” (such as the chitin nanofiber hydrogels resulting from [81], biodegradable films of chitosan with acid-soluble collagen mentioned in [139], or chitooligosaccharides possessing antioxidant activity)and a “purified compound” being very similar to those of “industrial use”, “biomedical applications”, and “water treatment” regarding the future applications referred to in the analysed publications. However, all categories showed an increasing trend, meaning that market applications will improve, particularly food and biomedical applications which are expected to nearly quadruplicate. The increase in food applications may be due to the use of chitin/chitosan as a natural preservative or coating agent due to their recognized antimicrobial properties [87,88,89,90,110]. As for the biomedical applications of chitin/chitosan, these are expected to increase given the potential of these compounds in tissue engineering, wound healing, and drug delivery systems [105] and the high market value of biomedical engineering (USD ~240,000 million in 2022, with a CAGR of 12.3% from 2023 to 2029 [140]). These results are somehow in line with published data on the patentology of chitin/chitosan, where 539 patent families covering chitin and its applications [116] and 3650 patent families covering chitosan and its applications were described [115], and where a large part of them are referred to for biomedical applications, material sciences, and engineering in the case of chitin, or for chitosan, biomedical applications, together with medical, dental, pharmaceutical, or toilet purposes.

The application of chitin/chitosan and its derivatives in water treatment is also expected to increase, given that the antimicrobial, pollutant-binding capacity and flocculant activity of these bioactive compounds may replace fossil-based or other products which harm the environment in the treatment of water and wastewater [119,141]. The water treatment chemicals market was valued at USD 23.500 million in 2018 [141] and it is also expected to grow [16,93,142]. 

Similar to chitin/chitosan, collagen has been applied in a wide range of sectors from biomedical (highest number of mentions) to cosmetic, pharmaceutical, food applications, and industrial use. As observed for chitin/chitosan, the number of mentions of collagens’ current applications in the analysed publications was low for all sectors (always <5) and the number of publications mentioning a “purified compound” or “new derived compounds” was comparable to that of current applications. However, regarding the future of collagen usage, the number of mentions increases greatly for the biomedical, cosmetic, pharmaceutical, and nutritional applications, but it is either maintained or decreases in the newly derived compounds and purified compound sectors, respectively. This trend suggests that the transition from R&D to commercial applications might be faster for collagen than for chitin/chitosan.

In biomedical applications, the number of mentions is strikingly high (30), in agreement with the growth expected for the biomedical engineering market, which therefore represents an exceptional financial opportunity for collagen and its derivatives. Collagen-based biomaterials have been applied in tissue engineering [68,105] and bone regeneration [143,144]. The fact that the analysed publications refer to collagen biomedical applications more often than chitin/chitosan biomedical applications (30 vs. 10) suggests that the biomedical engineering market might present even more opportunities for collagen than for chitin/chitosan. Also worth exploring are the cosmetic and pharmaceutical applications of collagen, according to the publications analysed in the present study. Although collagen is widely used in antiaging and skincare products [41,105,145] and the global pharmaceutical and cosmetic market is the most valued (USD 1.69 billion in 2021, [146]) among the markets considered here, the current use of collagen in cosmetic and pharmaceutical applications was only mentioned in three publications, while its future applications were referred to in twelve. These numbers may indicate that this high-value market is difficult to venture into, despite the potential applications of marine-derived collagen and its derivatives and the high return on investment expected. Collagen bio-based materials also showed potential for nutritional applications, such as antioxidants and nutritional supplements [147,148].

Overall, chitin/chitosan and collagen, as well as their derivatives, are recognized for their potential to significantly advance biomedical, cosmetic, pharmaceutical, food, and other industries, but such applications and their contribution to long-term and innovative solutions is seldom documented in the scientific literature, suggesting that there are still important gaps in knowledge transfer between R&D in the academia and industrial applications, as well as between industrial R&D and the scientific community, which might be due to legal limitations imposed by the industry.

### 3.6. Trends in Data Distribution along Each Value Chain

The scientific publications related to the chitin/chitosan and collagen value chains analysed in the present study focused mostly on how these bioactive compounds and/or their derivatives were obtained rather than on the later steps of both value chains and market-related information. Interestingly, more publications mentioned [70,149,150,151] the origin of raw materials, inputs/feedstock, and the pre-treatment and pre-processing steps in the collagen value chain than in the chitin/chitosan value chain, but the opposite trend was found [65,110,152] for processing and product manufacturing, consumption, chain outputs, and the interaction between stakeholders. This switch may indicate that the focus of scientific publications related to the chitin/chitosan value chain is changing towards the final steps. The importance placed on the initial steps of both value chains can be attributed to their critical role in laying the groundwork for marine-derived collagen production, as well as ensuring quality standards and meeting industry requirements [89,90]. Because such steps are fundamental, many publications were expected to address them.

A high percentage (~60%) of publications also mentioned future perspectives for chitin/chitosan and collagen products [68,150,152,153], indicating an interest in developing new products based on these bioactive compounds and/or developing new applications for marine-derived collagen and chitin/chitosan products. This high percentage also reflects the forward-thinking of researchers working on marine-derived chitin/chitosan and collagen and the active pursuing of their full potential for innovative applications, markets niches, and opportunities. However, most publications only mentioned these innovations as possibilities [150,154], without developing further into which technology readiness level (TRL) the marine-derived chitin/chitosan and collagen products are at or which level they might be at in the near future. This may be due to a lack of engagement between researchers and industry partners/end-users, as well as not considering market demands and needs at the R&D and process development stages. This is in line with general observations from de Wit-de Vries et al. [155] in their extensive review of barriers and opportunities to improving the overall knowledge transfer ecosystem reality across many disciplines. It can also be due to scientific journals supporting publications that cover the first technology readiness levels (TRLs) rather than those covering the complete value chain. This is a hypothesis that finds some support in the low percentage of publications reporting patents, industry opportunities, industry constraints and challenges, profitability, the type of companies involved in marketing the products, and the end-consumers, evidencing that there are substantial obstacles in transitioning from marine-derived chitin/chitosan and collagen knowledge and application potential to the actual application of such products in commercially viable and market-ready products [155]. The difficulty in navigating regulatory frameworks, dealing with safety and efficacy concerns, and meeting consumer demands for product attributes, pricing, and accessibility is also reflected in this “potential vs. real” gap [156]. It is therefore crucial to promote closer industry–researcher engagement, promote further market integration of scientific data, and broaden publication practices to highlight the practical implications of research [157,158,159]. Cross-cutting publications from the lab to market can speed up this concept of integration into scientific research projects and foster wider and more efficient industry–academia jointly developed products [159]. Collaboration among stakeholders is therefore critical for bridging the gap described above [160], as well as for overcoming the standardisation/certification and packaging/distribution challenges. These collaborative efforts from multiple stakeholders might need to follow new models where information travels back and forth at each step of the value chain for targeted investments in additional research, regulatory support, and an in-depth grasp of consumer preferences which successfully deliver innovative, sustainable, and market-ready products to end-users [161].

### 3.7. State-of-the-Art and Expressed Trends in the Chitin/Chitosan and Collagen Value Chains

The SWOT analysis (Table 1) performed for each value chain, considering the information provided in the analysed peer-reviewed scientific publications in order to evaluate the statuses and future scenarios of both value chains, revealed that one of the immediate strengths supported by this systematic review for each value chain is the high percentage (>80%) of scientific information which is published in highly scored (Q1 and Q2) journals. This finding indicates that the extensive knowledge on chitin/chitosan and collagen and their applications and potential is perceived as timely, sound, and relevant. In the same context, the well-documented processes used to obtain chitin/chitosan and collagen ensure consistency and standardisation, as well as replication and optimization, resulting in increased production efficiency and efficacy, thus adding to the strength of both value chains. It is important to note that the marine origin of the chitin/chitosan and collagen here considered is a significant strength as it addresses dietary and cultural restrictions that often apply to land-based and animal-based counterparts, thus expanding its market potential. This advantage results from avoiding religious food prohibitions like halal and Hindu dietary regulations [162,163]. Based on the information provided by the scientific publications on the chitin/chitosan value chain analysed, cost-efficient and environmentally friendly methods are being used to improve extraction techniques. Moreover, raw materials from the food processing industry are also being used, contributing to a more sustainable and circular economy. Environmentally friendly practices, such as the reuse of waste materials, are also in use for obtaining collagen. Such sustainable practices help reduce the environmental impact of the chitin/chitosan and collagen industries while promoting resource efficiency and can therefore be key differentiator factors. In addition, the search for novel, sustainable extraction techniques aligns with current sustainability goals which emphasize minimizing resource consumption and environmental harm [122,164]. Chitin and its derivatives have already found applications in advanced biomaterials due to their unique properties [12]. They can contribute significantly to a variety of industries thanks to their versatility [12]. The use of collagen-based antioxidants, which are non-toxic and offer nutritional benefits, further enhances its value [165]. Notably, the strategy of converting marine food waste into value-added products is in line with efforts made around the world to mitigate ecological and economic imbalances brought on by marine waste, directly contributing to the achievement of sustainability goals [12,122].

The value chains for chitin/chitosan and collagen, however, have a few weaknesses. Both value chains presented a lack of scientific studies considering macroeconomic factors, such as market trends and economy fluctuations, which limits the industries’ ability to adapt to changing conditions and capitalise on opportunities. Furthermore, social sustainability mentions were rare in the analysed publications, which raises concerns on how the chitin/chitosan and collagen industries deal with their own social context and with those of the industries they relate with (e.g., fisheries, aquaculture, food processing industry). According to UNICEF or UN reports, to ensure long-term sustainability, it is critical to address labour conditions, community engagement, and ethical practices. Although relying on raw materials sourced from the food processing industry increases the environmental and economic sustainability of the chitin/chitosan industry, it also means this industry is vulnerable to fluctuations in the supply of such raw materials. Contrastingly, the lower utilisation of raw materials from the food processing industry in the collagen industry than in the chitin/chitosan industry suggests that opportunities for improving resource use efficiency and waste reduction might be missed in the former. Sustainability issues are raised by the current extraction methods’ environmental impact [122]. Traditional extraction methods might not be able to fulfil today’s standards for sustainability, raising questions about their effects on the environment and their use of resources [58]. The adaptability of chitin/chitosan and collagen from newly explored marine sources to diverse markets remains uncertain due to factors such as variations in quality and scalability and the need for market-specific certifications, such as the high-tech applications. Its commercialisation may be hindered by the public’s view of the source, which may generate doubt on the quality and safety of the product [166]. Another weakness identified in the collagen value chain is the decreasing tendency observed in the current vs. future mentions to newly derived compounds and purified collagen products, suggesting that there might be a deceleration in collagen innovation, which may lead to missing new applications and new business opportunities.

The opportunities for chitin/chitosan and collagen are substantial. As a direct result of the rising demand for sustainable and alternative food sources, their potential applications could be used in a variety of industries, including biomedical, food, industrial, and water treatment [12,167]. In the analysis here performed, several publications referred to the anticipated expansion of biomedical, cosmetic, pharmaceutical, and food applications, as well as the industrial use of marine-derived chitin/chitosan and collagen. Diversification and entry into new markets are made possible by broadening the range of applications. Both value chains also have opportunities in animal supplements and nutritional applications, capitalising on the growing demand for well-being and health products for both human and veterinary markets. On the other hand, it is worth noting that the use of fishing discards or fisheries’ waste material reported in some of the analysed publications, which are also a large trend in the sector, may provide interesting models for coastal and fishing communities. By giving fishermen a new possible stream of income and by decreasing waste, using waste materials for chitin/chitosan and collagen extraction handles concerns of social equity and promotes a circular economy [168]. Unlike collagen from land animals, marine-derived collagen reduces the risk of disease transmission and religious concerns, potentially opening new markets [122].

The chitin/chitosan and collagen industries face several threats in addition to the identified opportunities. Growth may be hindered by concerns with chitin/chitosan extraction methods’ sustainability and quality, as well as by issues with the high cost of production and storage [12,40,169]. To guarantee the safety of the products, issues like microbial or viral contamination, which are often connected to products derived from animals, still pose a concern. The large quantities of food processing waste and fishing by-catches discarded, which could otherwise be used for the extraction of these compounds, contribute significantly to environmental pollution, while risking human health and the fishing industry’s sustainability [170,171]. For this reason, this source of raw material can also be seen as a threatened one, as it tends to disappear as more regulation and zero-waste focused strategies are being implemented. On the other hand, relying heavily on fisheries for the supply of needed raw material makes the industry susceptible to fluctuations in marine resources, which can be seen as incentivizing overfishing, and potential resource depletion. As researchers explore new biomass sources for chitin/chitosan and collagen, competition with food needs also increases, potentially effecting market dynamics. The sustainability and profitability of the value chains may be affected by the development of higher value uses from the same biomass resources, which might divert resources from the extraction and processing of chitin/chitosan and collagen. Another relevant aspect is that both value chains lack scientific literature focusing on future market trends, industry opportunities, and technological advancements. There is a lack of studies referencing patents, for example, which suggests that researchers tend to overlook intellectual property rights and opportunities. This is likely to limit the ability of both industries to adapt and capitalise on emerging opportunities. Furthermore, the insufficient examination of profitability beyond economic sustainability raises concerns about the long-term viability of both value chains. Comprehensive research which considers cost structures, market demand, and value chain dynamics is essential for making informed decisions, allocating resources, and providing additional value to scientific research.

**Table 1 marinedrugs-21-00605-t001:** SWOT analyses of the chitin/chitosan (orange) and collagen (blue) value chains [12,40,58,122,163,164,165,166,168,169,170,171,172,173].

SWOT	Chitin/Chitosan	Collagen
**Strengths**	A high percentage (>80%) of scientific information is published in highly scored (Q1 and Q2) journals. The literature well documents the extraction processes to ensure consistency and standardisation. Coming from marine sources overcomes current barriers to land-based and/or animal-based counterparts and diet restrictions existing worldwide (halal, Muslim, Hindu, etc.). New, more efficient and more environmentally friendly methods are documented by several authors. Current extracted chitin, and its derivatives, already have an important role as components of advanced biomaterials. In this era of climate change, the strategy of producing chitin from wastes and converting it to value-added products is highly valued to mitigate the ecological and economic imbalances due to marine food wastes.	A high percentage (>80%) of scientific information is published in highly scored (Q1 and Q2) journals. The literature well documents the extraction processes to ensure consistency and standardisation. Coming from marine sources overcomes current barriers to land-based and/or animal-based counterparts and diet restrictions existing worldwide (halal, Muslim, Hindu, etc.). New, more efficient and more environmentally friendly methods are documented by several authors. One of the applications of collagen, collagen-based antioxidants, are highly valued, because unlike synthetic antioxidants, collagen-based ones are non-toxic and can also supply nutritional benefits to consumers. The extraction of collagen from marine wastes such as discards, and side streams helps to achieve one of the goals of EU fishing policies by reducing post-harvest losses.
**Weaknesses**	The lack of scientific studies considering macroeconomic factors. Social sustainability seems to be ignored or not integrated into available published data. The degree of dependence from raw materials sourced from the food processing industry (subject to fluctuations). The chemical processes used to obtain chitosan during recent decades are considered to have a big environmental footprint and the resulting chitosan does not meet the requirements of high-tech applications. Many of these new sources from which chitin/chitosan derived have yet to be proven to be adaptable and usable in many different markets. The eco-friendly method of chitin/chitosan extraction does not achieve the levels of yield and purity of the chemical methods and is still in a lab-scale phase.	The lack of scientific studies considering macroeconomic factors. Social sustainability seems to be ignored or not integrated into available published data. Traditional protocols applied to the extraction of collagen are outdated, mainly with respect to present demands to develop more sustainable processes. Literature data suggest a decreasing innovation tendency in developing new compounds and purified collagen products. The public perception of the origin of the product (marine wastes) may hinder its commercialisation. The adaptability of this marine-derived collagen to penetrate highly regulated markets is yet to be proven.
**Opportunities**	There is a vast number of possible applications for chitin and chitosan, with special focus on biomedical applications, food, industrial use, water treatment, and new applications in nutritional products are being exploited. The use of waste/discards raw materials is a new way of improving social equality, as well providing another stream of income for fishermen. More conscious consumers demanding sustainable and alternative food sources can be appeased by the marine-derived chitin/chitosan.	There is a vast number of possible applications for collagen, with special focus on biomedical applications, food applications, industrial use, cosmetic and pharmaceutical applications, and new applications in nutritional products and supplements for animals. Major sources for commercial collagen are the skin and bone of land animals, such as pigs and cows, and these sources are heavily associated with the risk of transference of diseases or religious issues; marine-originated collagen can help to tackle these challenges. More conscious consumers demanding sustainable and alternative food sources can be appeased by the marine-derived chitin/chitosan.
**Threats**	Current chitosan production methods and technologies experience a lack of quality in terms of potential purity and reproducibility, sustainability difficulties due to substantial pollutant emissions during the production process, or excessive production and storage costs. Challenges such as allergenic or viral contamination, normally related to animal originated products, are still to be properly addressed. The large quantities of food processing waste discarded could be used as a raw material for the extraction of chitin and may cause an enormous pollution problem. The high dependence on fisheries’ catches and supply is a risk. Many new sources of these products are being studied and competition is fierce. New applications with higher market values may be developed from waste and by-catch raw materials, making them competing uses of the same biomass. There is a disconnection between academic research outputs and market needs/applicability.	The constant discards of by-catches pose a serious threat to marine ecosystems, human health, and the sustainability and development of the fishing industries. Large quantities of food processing waste discarded could be used as a raw material for the extraction of collagen and may cause an enormous pollution problem. The high dependence on fisheries’ catches and supply is a risk. Many new sources of these products are being studied and competition is fierce. New applications with higher market values may be developed from waste and by-catch raw materials, making them competing uses of the same biomass. There is a disconnection between academic research outputs and market needs/applicability.

Both marine-originated chitin/chitosan and collagen value chains show strengths that make them promising biomaterials for diverse applications. Their potential for sustainability and capacity to overcome dietary limitations are in alignment with the rising demand for eco-friendly products. For them to succeed, it is essential to address their weaknesses, which include market-specific certification needs and environmental considerations taken up in new extraction processes. Moreover, the industries should carefully manage ecosystem impacts, competition, and the potential diversion of biomass resources to higher value applications. Ultimately, the future of these marine-derived biomaterials depends on strategic approaches that maximize opportunities while mitigating risks, demanding a joint approach between academia research and industry up-takers.

The PESTEL analysis here performed (Table 2) highlights several significant factors that currently influence or are expected to influence the sustainability and competitiveness of the chitin/chitosan and collagen value chains. Each factor has its own set of implications and sheds light on the intricate dynamics of the value chains.

The chitin/chitosan and collagen value chains’ policy context is a critical determinant. Government regulations, particularly import/export restrictions, have direct impacts on the availability and trade of raw materials and finished products, and marine conservation laws, such as the *Convention on Biological Diversity* (which entered into force on December 1993 and is currently signed by 168 parties; https://www.cbd.int/doc/legal/cbd-en.pdf accessed on 29 September 2023), the *OSPAR Convention* (which entered into force on March 1998 and is currently signed by 16 parties; https://www.ospar.org/site/assets/files/1169/ospar_convention.pdf accessed on 29 September 2023) and the *Marine Strategy Framework Directive* (Directive 2008/56/EC from June 2008; https://eur-lex.europa.eu/legal-content/EN/TXT/PDF/?uri=CELEX:32008L0056 accessed on 29 September 2023), ensure the use of sustainable raw material sources; health and safety regulations ensure product quality and traceability, minimizing the risks for consumers. Compliance with established rules is key for international trading while maintaining industry integrity. Initiatives that foster the circular bioeconomy, such as the *European Circular Bioeconomy Policy Initiative* (from January 2021; https://ecbpi.eu/wp-content/uploads/2021/02/ECBPI-manifesto.pdf accessed on 29 September 2023), accelerate the development of new and more efficient value chains, increasing the opportunities for these new value chains but also for the already established ones.

Economic factors also effect the performance of the chitin/chitosan and collagen value chains. Consumer preferences, lifestyle trends, and culture-driven market demand, which is a major driver for the growth of marine-derived chitin/chitosan and collagen industries, also benefit from the consumers’ interest on environmentally friendly and sustainable products [172]. The higher costs of labour, energy, and raw materials [172], as well as fluctuations in the availability of the latter caused by geopolitical conflicts, can have a significant impact on the profitability of both industries and their associated value chains. Currency exchange rates and high inflation can also effect the global competitiveness of the marine-derived chitin/chitosan and collagen industries [173]. Economic recessions or economic growth are of extreme importance for these value chains as well. Consuming patterns are influenced by the amount of available income that households possess [174], which may influence the demand for products within these value chains.

Product demand is heavily influenced by consumer preferences, including those related to culture, and therefore, social factors may either restrain or facilitate the availability and market growth of marine-derived chitin/chitosan and collagen products [162]. Because they originate mostly from materials that would otherwise be discarded, the consumers’ growing interest in sustainable, ethical, cruelty-free, and eco-friendly alternatives has created a positive market environment for these products [12,175]. Furthermore, lifestyle trends focusing on health and wellness have driven a high demand for collagen-based products in the cosmetic and nutraceutical industries based on the proven benefits of such products [175].The growing demand for marine food, due to the growing world population, is driving an increase in food waste that could be used as a raw material for the products within these value chains [176]. The aging population social pyramid demands new solutions regarding their well-being and quality of life, and these value chains can become extremely relevant for this, given the properties of chitin/chitosan and collagen [177].

Technological factors are key drivers of innovation and the development of the chitin/chitosan and collagen value chains. Advances in biotechnology and processing techniques have transformed extraction and purification methods, making them more efficient, clean, and sustainable [122,164]. The development of new methods has also increased the quality and purity of chitin/chitosan and collagen. Technological advancements have also led to new product development ideas and opportunities and have expanded chitin/chitosan and collagen applications in wound healing [150], tissue engineering [178], and drug delivery bio-based materials [100]. The raw material used for the extraction of both chitin/chitosan and collagen can influence its possible applications. Due to their marine origin, the presence of odour, taste, and colour in the chitin/chitosan and collagen final powders limit their applications in sectors such as cosmetic applications and food applications [175].

One of the major challenges for the chitin/chitosan and collagen value chains is climate change. Rising sea temperatures, ocean acidification, increased storm frequency, and habitat destruction all have huge impacts on the availability and biodiversity of the marine organisms used as raw materials for extracting chitin/chitosan and collagen [179]. Overfishing contributes further to depleting marine resources and disrupting ecosystems. Sustainable practices and responsible sourcing are therefore critical for the long-term viability of the chitin/chitosan and collagen value chains to alleviate the effects of these environmental factors. The contamination of the organisms that are the source of the raw material through pollution might impact the quality, purity, and safety of the extracted chitin/chitosan and collagen [180].

Finally, legal aspects and regulatory frameworks have a significant impact on the chitin/chitosan and collagen value chains. Intellectual property laws protect innovative technologies and product formulations, encouraging R&D. Biodiversity protection laws, benefit sharing arrangements, and the Nagoya protocol influence the prospects of new chitin/chitosan and collagen marine sources as well as the economic models into which they can participate and generate income. Product liability regulations ensure that safety standards are met to reduce the risks to consumers and to maintain industry integrity. Labour laws ensure that workers are treated fairly and that ethical practices are followed throughout the value chains. Adherence to the legal and regulatory frameworks is critical for stakeholders seeking to enter competitive and highly developed markets while protecting intellectual property rights, improving product quality, and promoting ethical and responsible behaviours along the entire chitin/chitosan and collagen value chains. Stricter laws and regulations can impact the performance of a company in terms of diverse factors such as productivity and profits, which, in return, will most likely effect the employees and their families [181].

Overall, the PESTEL analysis reflects the complex web of factors influencing the chitin/chitosan and collagen value chains, clearly reinforcing that these are better suited to be treated as value networks/webs, as mentioned above. The findings highlight the importance of government regulations, market demand, consumer preferences, technological advancements, environmental challenges, and legal frameworks in shaping these industries’ sustainability and competitiveness. Understanding and addressing these factors is critical for developing the industry and for engaging stakeholders, policy makers, and researchers in fostering sustainable practices, complying with regulations, and maintaining a competitive advantage in these evolving value chains. By proactively addressing these factors, these industries can maximise their potential while also protecting the environment and meeting societal needs.

**Table 2 marinedrugs-21-00605-t002:** PESTEL analysis.

PESTEL	
**Political** [182,183,184,185,186]	Government regulations, such as import/export restrictions, marine conservation laws, tariff policies and safety regulations, can affect the global market. Regulations on fishing practices and marine biodiversity conservation can limit raw materials’ availability. Government incentives or funding for sustainable marine resources may influence the availability and cost of raw materials. The current political drive and initiatives to foster circular bioeconomy are accelerating the development of new value chains and strengthening the logistics and opportunities in current ones.
**Economical** [172,173,174]	Market demand, production costs, and currency exchange rates can affect profitability and competitiveness. Price fluctuations in raw materials, such as fish skins or crustacean shells, can affect the availability and cost of chitin/chitosan and collagen. The instability of countries (due to political tensions, armed conflicts, wars, or economic crisis) that supply or consume the raw materials or the finished products can impact the pricing and the stability of the supply chain. Economic recessions or economic growth effect consumer spending patterns, and consequently, the demand for products within both value chains. More suitable and higher value market applications for the same raw materials can hinder or alter dramatically these value chains’ development from these marine sources (e.g., focus on new bioactive compounds)
**Social** [12,162,176,177]	Consumer preferences, lifestyle trends, and culture effect the demand for marine-derived products. The growing interest in sustainable, cruelty-free, ethical, and eco-friendly products may increase the demand for marine-derived chitin/chitosan and collagen products. The growing world population is driving the demand for more marine food, which leads to more raw material for these value chains. The aging population demands new solutions to improve their quality of life, and collagen and chitin/chitosan play relevant roles in many aspects of healthy lifestyles. Globalisation can play a role in shaping consumer behaviours, with trends established by online personalities able to increase demand for products related to health and well-being.
**Technological** [100,122,150,164,175,178]	Advances in biotechnology and processing techniques can improve production and processing efficiency, as well as the development of new products and applications. The use of advanced extraction and purification techniques has enabled the use collagen and chitin/chitosan in biomedical applications. The inability to use the marine-derived products in certain market applications exists due to a lack of desired characteristics (e.g., lack of odour or colour for cosmetic applications; lack of unpleasant taste or odour for food applications).
**Environmental** [179,180]	Climate change and resource depletion effect the availability and sustainability of chitin/chitosan and collagen sources. Overfishing, biodiversity loss, and habitat destruction impact raw materials’ availability. Pollution, such as plastic waste and chemical pollutants, can contaminate the organisms that are the source of the raw material, impacting the quality and safety of the extracted chitin/chitosan and collagen. The growing demand for raw materials and industrial production of these new products may cause new sources of pollution or environmental load.
**Legal** [181,187]	Intellectual property laws, product liability regulations, and labour laws impose restrictions throughout the chitin/chitosan and collagen value chains. Existing intellectual property landscape makes it harder to innovate for collagen or chitin/chitosan new molecules. Superiority and best-in-class may need to be developed as cases for highly regulated markets such as pharmaceutical or food application industries. Stricter regulations can impact businesses’ practices, jeopardizing employment and the consequent well-being of employees and their families.

## 4. Materials and Methods

### 4.1. Literature Search and Database Construction

A detailed and comprehensive literature search was conducted to systematise the available scientific information related to the chitin/chitosan and collagen value chains. Combinations of six to nine keywords, among the sixteen selected to represent the two value chains, were used to retrieve peer-reviewed scientific publications from Scopus and Web of Science datasets from 1954 to 2023 (Appendix A). The selection flow is presented in Figure 9. 

The keywords used in search combinations were as follows: Aquaculture; Bio* Waste; Chitin; Chitosan; Collagen; Collagen Hydrolysate; Crustaceans; Fisheries; Industr*; Marine; Marine Resources; Market; Market Demand; Return on Investment; Shellfish; Value Chain. All studies retrieved for each combination and involving chitin/chitosan and collagen of marine origin were considered.

This systematic review was structured according to the Preferred Reporting Items for Systematic Reviews and Meta-Analyses (PRISMA) guidelines [188]. 

The collection of scientific publications was performed using two databases: Web of Science (https://www.webofscience.com/) and Scopus (https://www.scopus.com/), both accessed on 1 February 2023. The search was first performed by topic (title, abstract, keywords) with no limitation of time span. This search query resulted in 1215 publications. After the removal of duplicates and publications that failed to meet the inclusion criteria, the title and abstract of 596 publications were screened. During this process, 11 publications were excluded because they were not available for the authors. Of the 471 full-text publications considered eligible, a final set of 218 was considered relevant for addressing the three research questions referred to in the Introduction section. These publications are listed in Appendix A.

To define the relevance of the publications, it was checked whether the information in the abstract and conclusions fitted the inclusion criteria. The authors then analysed the Material and Methods section to confirm the results. Only information present in the publication was extracted to guarantee that the results’ interpretation was as objective as possible. In cases of disagreement between the two authors, a third author addressed the issue to decide whether the publication should be included or not. The diverse backgrounds of the authors allowed for a detailed interpretation of the data and for a reduction in the possibility of missing significant information.

### 4.2. Inclusion Criteria and Data Extraction

Although this systematic review is focused on the performance of products derived from chitin/chitosan and collagen, studies not including market-related information were also considered if they contained information on stakeholder interactions within each value chain. Concerning market performance, studies analysing the current market status (e.g., current players, market size, market volatility) and/or including projections of market behaviour (e.g., growth rates, increase in investors’ interest) were also included.

As only first-hand information was considered, review publications were not included to avoid duplicating information or inserting other authors’ opinions. Extended abstracts, books and books chapters, conference summaries, and other non-peer-reviewed literature were not included in the database. Non-English publications were also excluded.

### 4.3. Data Analyses

To characterise the source of information, the selected 218 publications were analysed for quartile (Q) information, via SCImago (https://www.scimagojr.com/ accessed on 1 February 2023), and a Q analysis of the publications was performed for each value chain, as well as for the country of the corresponding author. The best quartile attributed to the journal for the corresponding year was selected. If no quartile was assigned to the journal that year, then the last available quartile before the publication year was assumed. If no quartile was available, Q4 was assumed. In cases where the publication was published online before being published by the journal, the most favourable quartile was selected. The country of the corresponding author(s) was considered that of the institution/organisation where the research took place. If there was more than one corresponding author, or if the authors were conducting research in more than one country, all countries were considered. Concerning the origin of raw materials, “local market”, “collected”, and “by-catch” were considered as “Fisheries”. When a “commercial” source was indicated, this was considered as “Undisclosed”. Finally, “crab shells” were considered to originate from the “Food processing industry”.

To address the research questions formulated in the present study, the 218 publications selected were analysed according to the evidence provided in the following contexts: (i) value chain, including the flow between the eight drivers of change (raw material origin, inputs/feedstock, pre-treatment/pre-processing, processing and product manufacturing, standardisation/certification, packaging/distribution, consumption, and chain outputs); (ii) sustainability, considering environmental, socioeconomic, and circular economy perspectives; (iii) market-related information, namely, business models and return on investment; and (iv) any lessons learned or recommendations that might support future perspectives for this blue economy sector.

## 5. Conclusions

This systematic literature review offers insightful information about the scientific knowledge gap in the chitin/chitosan and collagen value chains, mostly located at the product application level. Despite suggesting several current and potential applications of both chitin/chitosan and collagen in biomedical, pharmaceutical, cosmetic, and food industries, and as viable alternatives for replacing chemicals in wastewater and water treatment, scientific publications rarely address the success of such applications nor their market or economic value. Nevertheless, the market value of chitin/chitosan and collagen has been addressed in several reports and thus future research should include such reports to provide a more accurate picture of both value chains. Moreover, given the considerable contribution of China and India to the research on chitin/chitosan and collagen, scientific publications published in other languages than English should be considered to ensure that relevant information is not being missed. To further narrow the detected gap, the communication between stakeholders in the chitin/chitosan and collagen value chains needs to be fostered, particularly in the anticipated scenario of increased market value and application diversification of both chitin/chitosan and collagen products. The identification of new uses for these compounds and their derivatives by the research community will fuel such growth and should be taken by the industry as an opportunity to establish future strategies well ahead of time and account for consumer demands, particularly those framed by their cultural background and sustainability concerns. Simultaneously, the industry should communicate its needs to the research community to facilitate the successful translation of scientific developments into commercial applications. Industry reverse pitching to academia must be fostered to close this gap. This approach should translate into investigations being more focused towards products that fulfil the needs of the industry and consumers, while being more sustainable economically, environmentally, and socially. Given the current dependence on aquaculture and fisheries to source raw materials for chitin/chitosan and collagen value chains, whose production is mostly secured by Asian and Latin American countries that are often poorer than those where most consumers of these value-added products originate from, the three dimensions of sustainability need to be evaluated in both value chains in light of this imbalance. Scientific publications should therefore foster transparency and compliance in their accepted publications in order to contribute to standard practices and fair and just procedures for all. Policy makers must engage with both academic and industry communities when designing new legal and funding frameworks to align the needs and incentives with value chain bottlenecks but also to ensure that negative impacts on the environment and on the health and social well-being of consumers are minimized.

## Figures and Tables

**Figure 1 marinedrugs-21-00605-f001:**
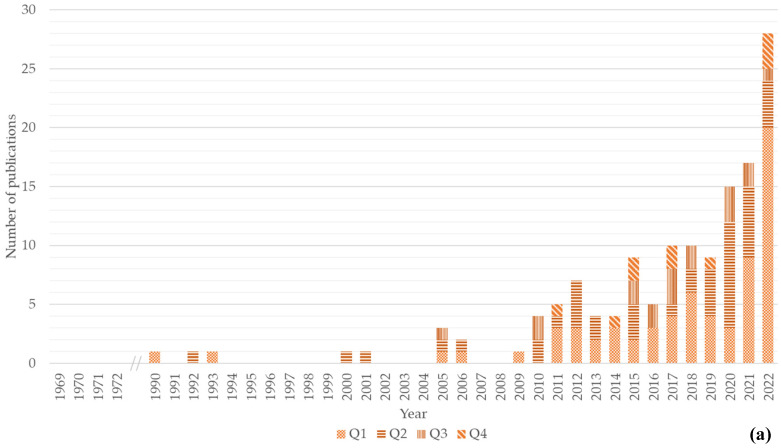
Number of publications per year and quartile (Q1–Q4). (**a**) Number of publications for the chitin/chitosan value chain; (**b**) number of publications for the collagen value chain. Quartile classification according to SCImago.

**Figure 2 marinedrugs-21-00605-f002:**
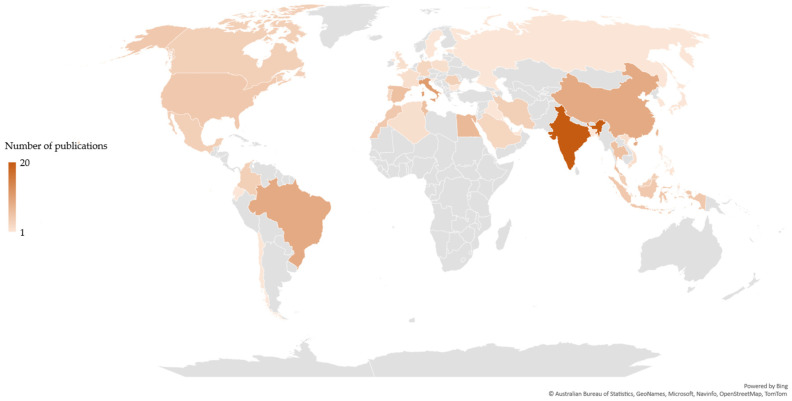
Geographic distribution of the chitin/chitosan value chain-related publications based on the country of the corresponding author(s).

**Figure 3 marinedrugs-21-00605-f003:**
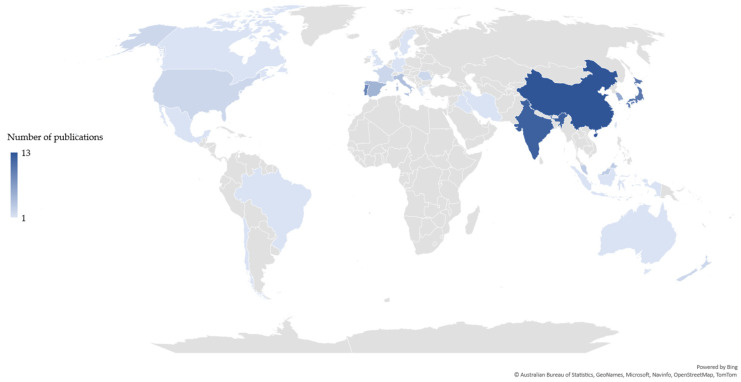
Geographic distribution of the collagen value chain-related publications based on the country of the corresponding author(s).

**Figure 4 marinedrugs-21-00605-f004:**
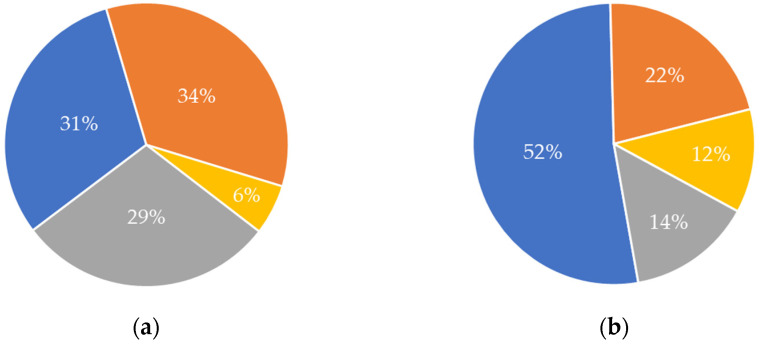
Origin of the raw materials as their frequency of occurrence in the publications analysed for each value chain. (**a**) Chitin/chitosan value chain; (**b**) collagen value chain. Blue, fisheries; orange, food processing industry; yellow, aquaculture; grey, undisclosed.

**Figure 5 marinedrugs-21-00605-f005:**
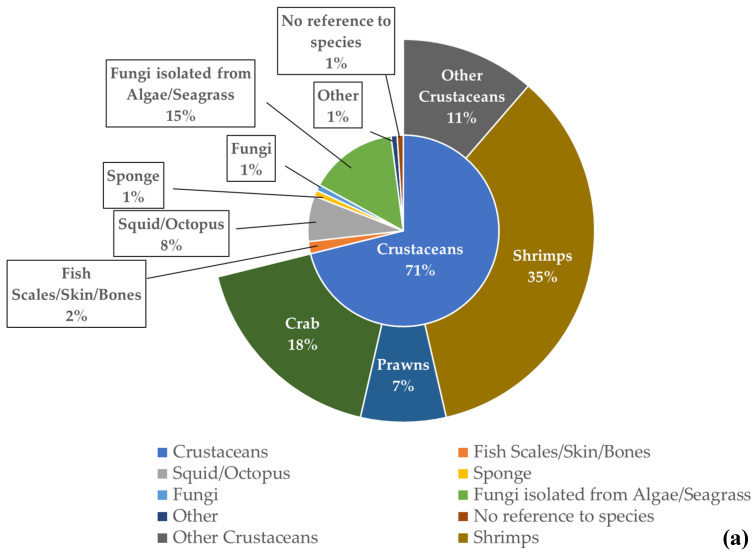
Feedstock used as source of extraction and their frequency of occurrence in the publications analysed for each value chain. (**a**) Chitin/chitosan value chain; (**b**) collagen value chain.

**Figure 6 marinedrugs-21-00605-f006:**
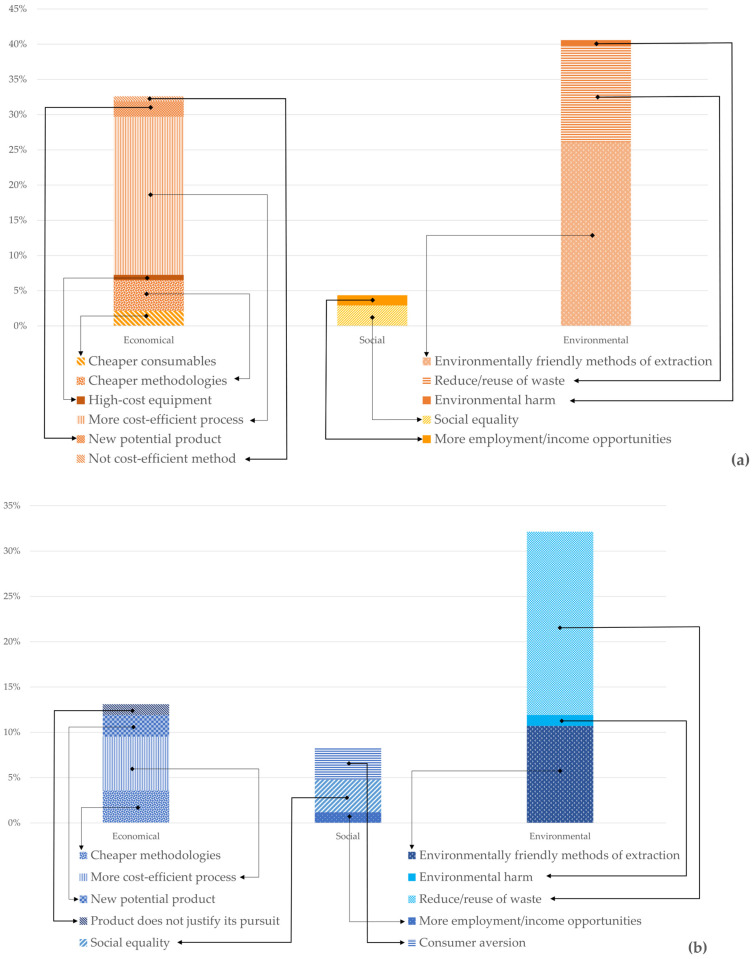
Percentage of publications referring to each of the three categories of sustainable practices per value chain. (**a**) Chitin/chitosan value chain; (**b**) collagen value chain. Economical mentions refer to cheaper consumables; cheaper methodologies; high cost equipment; more cost-efficient process; new potential product; not cost-efficient method; product that does not justify its use. Sustainability mentions refer to consumer aversion; more employment/income opportunities; social equality. Environmental mentions refer to environmentally friendly methods of extraction; environmental harm; reduce/reuse of waste.

**Figure 7 marinedrugs-21-00605-f007:**
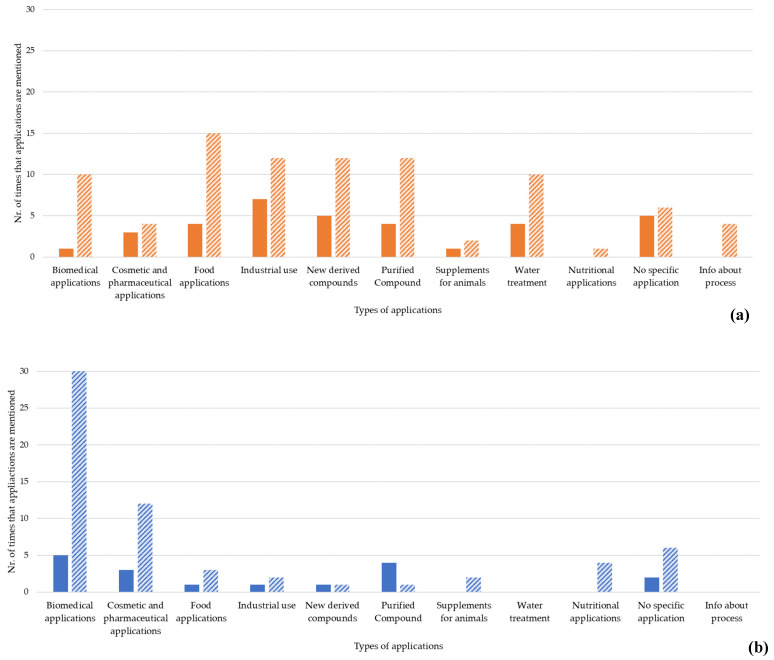
Number of current (block colour) and future (striped pattern) applications reported by sectors for chitin/chitosan (**a**) and collagen (**b**) products. The resulting bars correspond to the exact number of each application field mentioned as *current* or *future* applications in the analysed publications dataset.

**Figure 8 marinedrugs-21-00605-f008:**
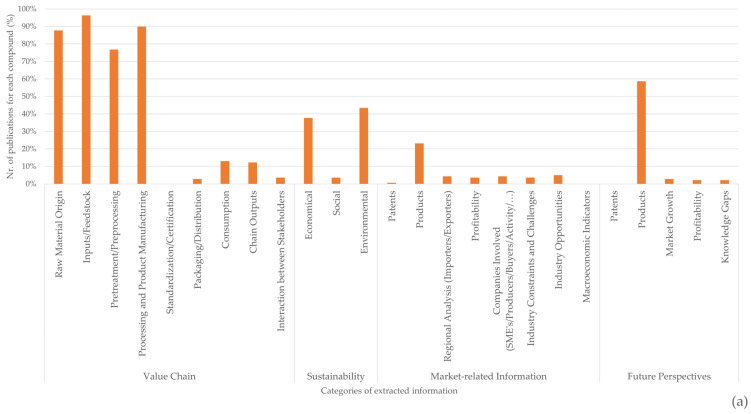
Percentage of publications referring to each category of extracted information for chitin/chitosan (**a**) and collagen (**b**) value chains.

**Figure 9 marinedrugs-21-00605-f009:**
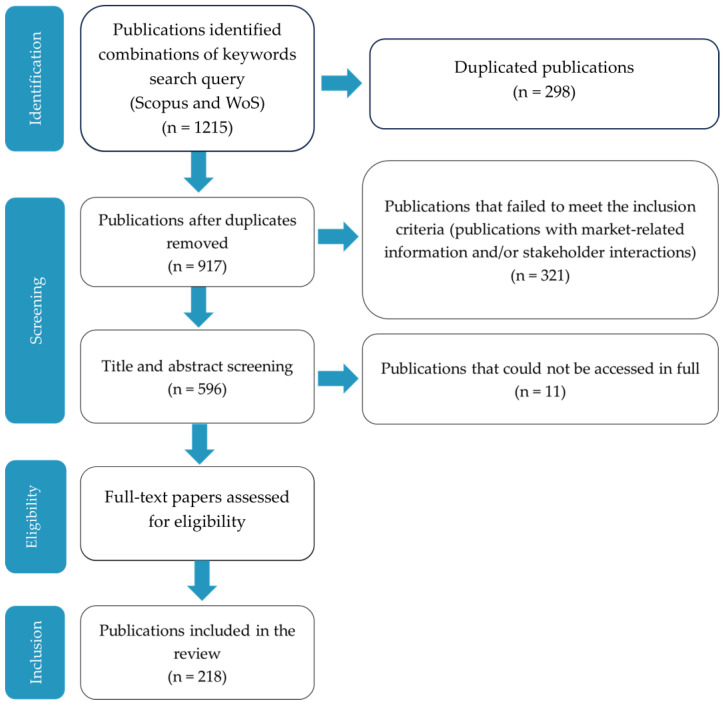
Publications’ selection process following PRISMA guidelines [188].

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
