# Peer review of "Current and Expected Trends for the Marine Chitin/Chitosan and Collagen Value Chains"

_marinedrugs, 2023, doi:10.3390/md21120605_

Round 1

Reviewer 1 Report

Comments and Suggestions for Authors

This is rather unusual review paper on chitin and collagen as fundamental structural biopolymers under study. However, such kind of information is necessary and will be used by broad diversity of experts and strudents. I can recommend this manuscript for publication after major revision.

Critical comments:

-          Surprisingly, you have overlooked two fundamental papers on chitin and chitosan entitled Kertmen A., Ehrlich H. (2022) Patentology of chitinous biomaterials. Part I: Chitin Carbohydrate Polymers 292, 119102; Kertmen A., Dziedzic I., Ehrlich H. (2023) Patentology of chitinous biomaterials. Part II: Chitosan. Carbohydrate Polymers Vol 301, Part A, 120224d. Both must be discussed in your work.

 -          Page 3, Line 109-110: You wrote: „Chitin is one of the most abundant biopolymers in

nature [20].“ Additional reference should be placed here: Tsurkan et al (2021). Progress in Chitin Analytics. Carbohydrate Polymers, 117204.

-          Page 3,Line 113: following reference must be inserted: Kaya et al (2017) On chemistry of γ–chitin. Carbohydrate Polymers 176:177–186

-          Electrochemical methods for chitin isolation from marine sources has been overlooked. Recommended references: Nowacki et al (2020) Electrochemical method for isolation of chitinous 3D scaffolds from cultivated Aplysina aerophoba marine demosponge and its biomimetic application. Applied Physics A 126:368; Nowacki et al (2020) Electrochemical approach for isolation of chitin from the skeleton of the black coral Cirrhipathes sp. (Antipatharia). Marine Drugs 18(6):297; Nowacki et al (2022) Electrolysis as a Universal Approach for Isolation of Diverse Chitin Scaffolds from Selected Marine Demosponges. Marine Drugs Vol. 20(11), 665

-          Both chitin and collagen represent unified templates for biomineralization and skeletogenesis. Insert this fundamental point into the text. See: Ehrlich H.(2010) Chitin and collagen as universal and alternative templates in biomineralization. International Geology Review 52:661–699

-          Also, you have overlooked that both structural biopolymers represent examples of such modern direction as “scaffolding strategy”. It means how to use naturally occurring 3D scaffolds made of chitin and collagen ( i.e. in sponges) in tissue engineering and technology. See: Tsurkan et al (2020) Modern scaffolding strategies based on naturally pre-fabricated 3D biomaterials of poriferan origin. Applied Physics A 126:382; Khrunyk et al (2020) Progress in Modern Marine Biomaterials Research. Mar. Drugs 18, 589; Dziedzic, et al. The Loss of Structural Integrity of 3D Chitin Scaffolds from Aplysina aerophoba Marine Demosponge after Treatment with LiOH. Mar. Drugs 2023, 21, 334 

-          Marine sponges represent examples of renewable sources of both chitin and collagens due to their large scale cultivation under marine ranching conditions. This direction shoud be briefly discussed in your review, too. See: Ehrlich et al (2018) Collagens of poriferan origin. Marine Drugs 16:79; Pozzolini et al (2021) Potential Biomedical Applications of Collagen Filaments derived from the Marine Demosponges Ircinia oros (Schmidt, 1864) and Sarcotragus foetidus (Schmidt, 1862) Mar. Drugs, 19(10), 563;

-          Chitin due to its thermostability up to 360°C found application in modern materials science within the framework of extreme biomimetics. This direction should be also represented in your manuscript. For details see: Wysokowski et al (2015) Chitin as a versatile template for extreme biomimetics. Polymers, 7:235–265; Ehrlich H., Wysokowski M., Jesionowski T. (2022) The Philosophy of Extreme Biomimetics. Sustainable Materials and Technologies Vol 32 e00447

Author Response

Vieira et al., 2023 – marinedrugs-2709662

Reply to REVIEWER 1 comments

We would like to thank the Reviewer 1 for his/her very relevant and complete comments that allowed us to reflect upon and further enrich our manuscript.

We have prepared a revised manuscript version with track changes and comment by comment replies to each reviewer’s points, as we believe it facilitates locating each alteration.

Additionally, we provide here a summary description of our actions towards each point Reviewer 1 has raised.

 Reviewer 1 comments in italics and each reply is presented next:

“This is rather unusual review paper on chitin and collagen as fundamental structural biopolymers under study. However, such kind of information is necessary and will be used by broad diversity of experts and strudents. I can recommend this manuscript for publication after major revision.

 Reply: We wish to thank the reviewer for acknowledging the broad scope and interest this manuscript will raise and it was precisely our goal when designing this approach. We have now significantly improved the manuscript with all your comments and have also increased our references from 106 originally to 190 references in the new version.

Critical comments:

- Surprisingly, you have overlooked two fundamental papers on chitin and chitosan entitled Kertmen A., Ehrlich H. (2022) Patentology of chitinous biomaterials. Part I: Chitin Carbohydrate Polymers 292, 119102; Kertmen A., Dziedzic I., Ehrlich H. (2023) Patentology of chitinous biomaterials. Part II: Chitosan. Carbohydrate Polymers Vol 301, Part A, 120224d. Both must be discussed in your work.

Reply: We wish to thank the reviewer for bringing our attention to these two relevant references. We were aware of them and have now included them. References to these patent studies were included in sections 3.2 and 3.5. We avoided more refereeing as we are analyzing the same aspects from an IP perspective on a different, yet unpublished, manuscript.  

- Page 3, Line 109-110: You wrote: „Chitin is one of the most abundant biopolymers in nature [20].“ Additional reference should be placed here: Tsurkan et al (2021). Progress in Chitin Analytics. Carbohydrate Polymers, 117204

Reply: We have mentioned this aspect and added this reference in the introduction

- Page 3,Line 113: following reference must be inserted: Kaya et al (2017) On chemistry of γ–chitin. Carbohydrate Polymers 176:177–186

Reply: This reference was added.

- Electrochemical methods for chitin isolation from marine sources has been overlooked. Recommended references: Nowacki et al (2020) Electrochemical method for isolation of chitinous 3D scaffolds from cultivated Aplysina aerophoba marine demosponge and its biomimetic application. Applied Physics A 126:368; Nowacki et al (2020) Electrochemical approach for isolation of chitin from the skeleton of the black coral Cirrhipathes sp. (Antipatharia). Marine Drugs 18(6):297; Nowacki et al (2022) Electrolysis as a Universal Approach for Isolation of Diverse Chitin Scaffolds from Selected Marine Demosponges. Marine Drugs Vol. 20(11), 665

Reply: Both this concept and mentioned references were added to the introduction section.

- Both chitin and collagen represent unified templates for biomineralization and skeletogenesis. Insert this fundamental point into the text. See: Ehrlich H.(2010) Chitin and collagen as universal and alternative templates in biomineralization. International Geology Review 52:661–699

Reply: We have mentioned this aspect and added this reference in the introduction.

- Also, you have overlooked that both structural biopolymers represent examples of such modern direction as “scaffolding strategy”. It means how to use naturally occurring 3D scaffolds made of chitin and collagen ( i.e. in sponges) in tissue engineering and technology. See: Tsurkan et al (2020) Modern scaffolding strategies based on naturally pre-fabricated 3D biomaterials of poriferan origin. Applied Physics A 126:382; Khrunyk et al (2020) Progress in Modern Marine Biomaterials Research. Mar. Drugs 18, 589; Dziedzic, et al. The Loss of Structural Integrity of 3D Chitin Scaffolds from Aplysina aerophoba Marine Demosponge after Treatment with LiOH. Mar. Drugs 2023, 21, 334

Reply: A new mention to scaffolding strategy included in introduction and references added.

- Marine sponges represent examples of renewable sources of both chitin and collagens due to their large scale cultivation under marine ranching conditions. This direction shoud be briefly discussed in your review, too. See: Ehrlich et al (2018) Collagens of poriferan origin. Marine Drugs 16:79; Pozzolini et al (2021) Potential Biomedical Applications of Collagen Filaments derived from the Marine Demosponges Ircinia oros (Schmidt, 1864) and Sarcotragus foetidus (Schmidt, 1862) Mar. Drugs, 19(10), 563;

Reply: We have added a new reference for chitin obtained from Sponges - REF Cuizhu Sun, Zhenggang Wang, Hao Zheng, Lingyun Chen, Fengmin Li, Biodegradable and re-usable sponge materials made from chitin for efficient removal of microplastics, Journal of Hazardous Materials, Volume 420, 2021, 126599, ISSN 0304-3894, https://doi.org/10.1016j.jhazmat.2021.12659 . We have also added the references for collagen from sponges as requested.

- Chitin due to its thermostability up to 360°C found application in modern materials science within the framework of extreme biomimetics. This direction should be also represented in your manuscript. For details see: Wysokowski et al (2015) Chitin as a versatile template for extreme biomimetics. Polymers, 7:235–265; Ehrlich H., Wysokowski M., Jesionowski T. (2022) The Philosophy of Extreme Biomimetics. Sustainable Materials and Technologies Vol 32 e00447

Reply: A new mention to thermal stability of chitin and usage in materials science was included in introduction and references added.

We hope that the changes performed in the light of such comments is satisfactory to the reviewer and editor and that our manuscript can now be accepted.

With kind regards, and in the name of all authors,

Helena Vieira

Date of submission: 25 October 2023

Date of this review: 06 November 2023

Date of the authors reply: 11 November 2023

Reviewer 2 Report

Comments and Suggestions for Authors

The manuscript entitled “Current and expected trends for the marine chitin/chitosan and collagen value-chains” by Vieira et al., reviews the scientific knowledge and publication trends along the marine chitin/chitosan and collagen value-chains and assesses how researchers, industry players, and end-users can bridge the gap between scientific understanding and industrial applications. The review is very interesting and well-organized and stresses the importance of the successful communication between the academic sector and the industry.

Nevertheless, there is a question that needs to be answered. Why did not the authors use the keyword Gelatin along with Collagen hydrolysate OR Collagen? Gelatin is the denaturation product of collagen that retains many of its activities and thus is used in almost the same applications as collagen. Moreover, since collagen isolation requires low temperatures, fish wastes are mostly used to produce gelatin and not collagen, also due to economical issues. Please explain and revise accordingly.

Some more points that have to be taken into consideration:

Figure 7 shows the reported future applications for chitin/chitosan and collagen products. How exactly are these numbers calculated? Please specify.

Lines 288, 296: Please correct to Figure 8.

S_Table 3: The references are written in alphabetical order only up to page 36. Please check.

Author Response

Vieira et al., 2023 – marinedrugs-2709662

 Reply to REVIEWER 2 comments

We would like to thank the Reviewer 2 for his/her very relevant and complete comments that allowed us to reflect upon and further enrich our manuscript.

We have prepared a revised manuscript version with track changes and comment by comment replies to each reviewer’s points, as we believe it facilitates locating each alteration.

Additionally, we provide here a summary description of our actions towards each point Reviewer 2 has raised.

 Reviewer 2 comments in italics and each reply is presented next:

“The manuscript entitled “Current and expected trends for the marine chitin/chitosan and collagen value-chains” by Vieira et al., reviews the scientific knowledge and publication trends along the marine chitin/chitosan and collagen value-chains and assesses how researchers, industry players, and end-users can bridge the gap between scientific understanding and industrial applications. The review is very interesting and well-organized and stresses the importance of the successful communication between the

academic sector and the industry.”

 Reply: We wish to thank the reviewer for his/her kind words on the relevance and content of this manuscript and recognizing it will raise interest to a broad range of stakeholders as it was precisely our goal when designing this approach. We have now significantly improved the manuscript with all your comments and have also increased our references from 106 originally to 190 references in the new version.

Nevertheless, there is a question that needs to be answered. Why did not the authors use the keyword Gelatin along with Collagen hydrolysate OR Collagen? Gelatin is the denaturation product of collagen that retains many of its activities and thus is used in almost the same applications as collagen. Moreover, since collagen isolation requires low temperatures, fish wastes are mostly used to produce gelatin and not collagen, also due to economical issues. Please explain and revise accordingly.

Reply: The focus of this paper was indeed on the Chitin/Chitosan and Collagen (pure compound and derivatives) value chains. The authors agree with Reviewer 2 comment that gelatin is indeed a high volume and easier to produce product when marine/fish derived raw materials are used. However, Collagen is a higher value product with specific high value applications like medical or pharmaceutical, and therefore we have chosen to focus the analysis on this keyword. For this reason, we would like to maintain the focus of the paper as this was indeed the keyword used and not gelatin. Additionally, using keyword gelatin would imply a completely new and different analysis of the whole dataset of literature not possible at this stage.

 Some more points that have to be taken into consideration:

Figure 7 shows the reported future applications for chitin/chitosan and collagen products. How exactly are these numbers calculated? Please specify.

Reply: Figure 7 represents exactly the number of mentions for each future application cited by the authors in all publications. For clarification the Y axis title was changed for “nº of times application is mentioned” and the following sentence “The resulting bars correspond to the exact number each application field is mentioned as current or future application in the analysed publications dataset” was added to the legend of the figure for clarification. A new figure 7 was also submitted.

Lines 288, 296: Please correct to Figure 8.

Reply: Thank you for point this out. It has been corrected now.

S_Table 3: The references are written in alphabetical order only up to page 36. Please check

Reply: Thank you for point this out. It has been corrected and a new S_Table 3 was submitted.

We hope that the changes performed in the light of such comments is satisfactory to the reviewer and editor and that our manuscript can now be accepted.

With kind regards, and in the name of all authors,

Helena Vieira

Date of submission: 25 October 2023

Date of this review: 07 November 2023

Date of the authors reply: 11 November 2023

Reviewer 3 Report

Comments and Suggestions for Authors

Chitin and its derivatives such as chitosan, together with collagen, are two biopolymers that in recent years have generated great interest in different industries due to their great properties such as antioxidants, biocompatible and non-toxic, among others. In this article, both scientific articles and their industrial applications are reviewed in order to unite both knowledge in the near future creating a more sustainable and circular economy.

The topic is interesting, but there are several aspects that should be improved:

- First, in my opinion there are too many authors in this review for the length of the manuscript.

- Introduction: Some more references to the year 2022 and 2023 with data from these years should be included.

- Results: It would be appreciated if a sentence could be added with the databases used, although they are included in the supplementary with more extension.

- Section 2.1: this section is very general; some example or more specific data should be added.

- Sections 2.4; 2.5 and 2.6: There are hardly any references, the text is too general. References and examples should be added.

- Line 313: Due to the fact that this article is intended to be published this year. The references should be updated and some of 2022 and 2023 should be added with concrete data.

- Line 448: When referring to cost efficiency, it is a very general term, examples should be given to know what it is based on and what it is talking about.

- When talking in the text about improving yields or making processes more environmentally friendly, again it is very general and references and examples should be given to justify those facts. how do they obtain them?

- Line 491: when talking about new derivative compounds, what are they?

- Sections 3.5 and 3.6: Good ideas and arguments, but there are no references in the text to know what the authors have based on, it seems to be an opinion and not a literature review. References are needed.

- Table 2. In the PESTEL, what references have been used.

- Figures: The letters of the axes should be increased.

In general it is an interesting topic, but I think it needs more depth in general throughout the text, more references and more concrete examples to justify what they have been based on when presenting all the results. For all these reasons, I think that a major revision should be necessary in order to publish it.

Author Response

Vieira et al., 2023 – marinedrugs-2709662

Reply to REVIEWER 3 comments 

We would like to thank the Reviewer 3 for his/her very relevant and complete comments that allowed us to reflect upon and further enrich our manuscript.

We have prepared a revised manuscript version with track changes and comment by comment replies to each reviewer’s points, as we believe it facilitates locating each alteration.

Additionally, we provide here a summary description of our actions towards each point Reviewer 3 has raised.

 Reviewer 3 comments in italics and each reply is presented next:

“Chitin and its derivatives such as chitosan, together with collagen, are two biopolymers that in recent years have generated great interest in different industries due to their great properties such as antioxidants, biocompatible and non-toxic, among others. In this article, both scientific articles and their industrial applications are reviewed in order to unite both knowledge in the near future creating a more sustainable and circular economy. The topic is interesting, but there are several aspects that should be improved.”

 Reply: We wish to thank the reviewer for his/her kind words on the relevance of this manuscript and recognizing it will raise interest to a broad range of stakeholders as it was precisely our goal when designing this approach. We have now significantly improved the manuscript with all your comments.

- First, in my opinion there are too many authors in this review for the length of the manuscript.

Reply: It is not clear to the authors the reason for this affirmation from the reviewer 3. This is a 39 page review article with supplementary material, and the authors referred herein have all contributed significantly for this analysis, either by actively engaging in detailed systematic bibliography reviewing, or by writing, analysing, drafting and discussion of the results. Therefore, all authors are relevant as explained in authors contributions in page 28 of this manuscript.

- Introduction: Some more references to the year 2022 and 2023 with data from these years should be included.

Reply: The original version contained 106 references many from 2021, 2022 and 2023. However, we would like to thank the reviewers for their precious inputs and comments, and we have now increased our references from 106 originally to 190 references in the new version, many also from recent years.

Results: It would be appreciated if a sentence could be added with the databases used, although they are included in the supplementary with more extension.

Reply: Indeed that information is not only on the supplementary materials tables, but also on the materials and methods component (see point 4.1 please).

- Section 2.1: this section is very general; some example or more specific data should be added.

Reply: This is a pure analytical section, where the authors simply described the obtained results in terms of number, impact and temporal distribution of gathered publications from the systematic literature search. This is a relevant result analysed further in the discussion to infere which value chain has been raising more interest from scientific community, which is more consolidated in terms of scientific outputs etc. We do not see what type of further examples can be added here.

- Sections 2.4; 2.5 and 2.6: There are hardly any references, the text is too general. References and examples should be added.

Reply: Again, this is a results section, where we simply describe the obtained results from the systematic searches performed. However, we have densified these results in all three sections 2.4, 2.5 and 2.6, with some concrete examples of retrieved publications from our search, and added the related references to our list.

Reply: Thank you for point this out. It has been corrected and a new S_Table 3 was submitted.

- Line 313: Due to the fact that this article is intended to be published this year. The references should be updated and some of 2022 and 2023 should be added with concrete data.

Reply: This is now in line 345/346. Both original manuscript references, 28 and 45 (now 47 and 112), are global market reports of 2023. However, we have included additional detail, including a specific marine collagen market report and some more data (ref 113 - https://www.gminsights.com/industry-analysis/marine-collagen-market).

- Line 448: When referring to cost efficiency, it is a very general term, examples should be given to know what it is based on and what it is talking about.

Reply: This is now in lines 491 forward. We have now densified this section with more examples extracted from literature dataset we analysed and referenced the respective documents.

- When talking in the text about improving yields or making processes more environmentally friendly, again it is very general and references and examples should be given to justify those facts. how do they obtain them?

Reply: This is now in lines 491 forward. We have now added some example references from our search and specified the claimed means to improve yields or make processes more sustainable made by those authors such as alternatives to expensive enzyme usage as in ref. 71, 134 and 135 or less energy demanding processes as in ref 137.

- Line 491: when talking about new derivative compounds, what are they?

 Reply: This is now in lines 538 forward. hese are small chemical modifications such chitin nanofibers hydrogels resultant form the work by Mushi et al. (2016), biodegradable films of chitosan with acid-soluble collagen (Arias-Moscoso et al., 2011) or chitooligosaccharides possessing antioxidant activity.
- Sections 3.5 and 3.6: Good ideas and arguments, but there are no references in the text to know what the authors have based on, it seems to be an opinion and not a literature review. References are needed.

 Reply: We would like to reinforce that indeed this is the discussion component of this review, and in fact the authors do express their informed opinions on some of the observed trends. However, we do support our opinions in acknowledged and peer reviewed publications. Section 3.5 had originally26 references cited and now has 30. In Section 3.6 indeed there were originally no cited references by mistake. We have now added 17 references to this section.  In particular we have also added the following text:

New Line 626 - This is in line with general observations from de Wit-de Vries, et al, 2018, in their extensive review of barriers and opportunities to improve the overall knowledge transfer ecosystem reality across many disciplines.

 - Table 2. In the PESTEL, what references have been used.

 Reply: We have now added 22 new references referring to several support documents to the oficial EU/national policies and strategies, EU directives and other relevant documents as publications ands reports.

 - Figures: The letters of the axes should be increased.

 Reply: We have now increase the contrast and size of letters and submitted new figures.

 In general it is an interesting topic, but I think it needs more depth in general throughout the text, more references and more concrete examples to justify what they have been based on when presenting all the results. For all these reasons, I think that a major revision should be necessary in order to publish it.

Reply: We thank you again for all your relevant points and have address them all, providing now what we believe is a more robust manuscript.

We hope that the changes performed in the light of such comments is satisfactory to the reviewer and editor and that our manuscript can now be accepted.

With kind regards, and in the name of all authors,

Helena Vieira

Date of submission: 25 October 2023

Date of this review: 06 November 2023

Date of the authors reply: 11 November 2023

Reviewer 4 Report

Comments and Suggestions for Authors

The authors offer a good and interesting new review of Current and expected trends for the marine chitin/chitosan and value collagen-chains. This work is new and interesting and will be well cited. The work is well written and logical, well illustrated with suitable pictures, diagrams, diagrams and graphs. The manuscript contains relevant references to contemporary literature and to classic works in the field. The review of the work available in the literature was carried out in compliance with modern methodological requirements; the authors also not only retell the literature, but also express their attitude to the problem and their critical analysis. I think this is very important. The authors' conclusions do not contradict previously published data in this area and are consistent with general patterns, and also correspond to the results obtained. However, I recommend a minor revision because I ask the authors to improve the quality of the numbers in many of the figures (they are currently hard to see). I also recommend strengthening the introduction with some literary references, for example DOI 10.1016/j.fpsl.2020.100534

Author Response

Vieira et al., 2023 – marinedrugs-2709662

 Reply to REVIEWER 4 comments

We would like to thank the Reviewer 4 for his/her very relevant and complete comments that allowed us to reflect upon and further enrich our manuscript.

We have prepared a revised manuscript version with track changes and comment by comment replies to each reviewer’s points, as we believe it facilitates locating each alteration.

Additionally, we provide here a summary description of our actions towards each point Reviewer 4 has raised.

 Reviewer 4 comments in italics and each reply is presented next:

“The authors offer a good and interesting new review of Current and expected trends for the marine chitin/chitosan and value collagen chains. This work is new and interesting and will be well cited. The work is well written and logical, well illustrated with suitable pictures, diagrams, diagrams and graphs. The manuscript contains relevant references to contemporary literature and to classic works in the field. The review of the work available in the literature was carried out in compliance with modern methodological requirements; the authors also not only retell the literature, but also express their attitude to the problem and their critical analysis. I think this is very important. The authors' conclusions do not contradict previously published data in this area and are consistent with general patterns, and also correspond to the results obtained. However, I recommend a minor revision because I ask the authors to improve the quality of the numbers in many of the figures (they are currently hard to see). I also recommend strengthening the introduction with some literary references, for example DOI 10.1016/j.fpsl.2020.100534.”

  Reply: We wish to thank the reviewer for his/her kind words on the relevance, content and format of this manuscript  and recognizing it will raise interest to a broad range of stakeholders as it was precisely our goal when designing this approach. We have now significantly improved the manuscript with all your comments and have also increased our references from 106 originally to 190 references in the new version, including the requested Reviewer 4 reference. We have also changed the figures font size and contrast to make them easier to read. Again, we thank you for your input and ee have now significantly improved the manuscript with all your comments.

 We hope that the changes performed in the light of such comments is satisfactory to the reviewer and editor and that our manuscript can now be accepted.

With kind regards, and in the name of all authors,

Helena Vieira

Date of submission: 25 October 2023

Date of this review: 31 October 2023

Date of the authors reply: 11 November 2023

Round 2

Reviewer 2 Report

Comments and Suggestions for Authors

The authors have taken into consideration all the reviewers' comments.

A minor comment

Please check the references, lines 1108-1110

Reviewer 3 Report

Comments and Suggestions for Authors

The authors have responded to the questions raised. They have modified many aspects of the text and added new references to improve the manuscript. Therefore, I think it should be published.